# Incidence, risk factors, and feto-maternal outcomes of inappropriate birth weight for gestational age among singleton live births in Qatar: A population-based study

**Salma Younes[1], Muthanna Samara[2], Noor Salama[3,4], Rana Al-jurf[5], Gheyath Nasrallah[6], Sawsan Al-Obaidly[7], Husam Salama[8], Tawa Olukade[8], Sara Hammuda[2], Ghassan Abdoh[8], Palli Valapila Abdulrouf[1,9], Thomas Farrell[1,7], Mai AlQubaisi[8], Hilal Al Rifai[8], Nader Al-Dewik[5,10,11,12]***

**1** Department of Research, Women's Wellness and Research Center, Hamad Medical Corporation, Doha, Qatar, **2** Department of Psychology, Kingston University London, Kingston upon Thames, United Kingdom, **3** Health Profession Awareness Program, Health Facilities Development, Hamad Medical Corporation (HMC), Doha, Qatar, **4** American University in Cairo (AUC), Cairo, Egypt, **5** College of Health and Life Science (CHLS), Hamad Bin Khalifa University (HBKU), Doha, Qatar, **6** Department of Biomedical Science, College of Health Sciences, Member of QU Health, Qatar University, Doha, Qatar, **7** Obstetrics and Gynecology Department, Women's Wellness and Research Center, Hamad Medical Corporation, Doha, Qatar, **8** Department of Pediatrics and Neonatology, Neonatal Intensive Care Unit, Newborn Screening Unit, Women's Wellness and Research Center, Hamad Medical Corporation, Doha, Qatar, **9** Department of Pharmacy, Women's Wellness and Research Center, Hamad Medical Corporation, Doha, Qatar, **10** Interim Translational Research Institute (iTRI), Hamad Medical Corporation (HMC), Doha, Qatar, **11** Faculty of Health and Social Care Sciences, Kingston University, St. George's University of London, London, United Kingdom, **12** Department of Pediatrics, Clinical and Metabolic Genetics, Hamad General Hospital, Hamad Medical Corporation, Doha, Qatar

* Naldewik@hamad.qa, Nader.Al-Dewik@kingston.ac.uk

**Data Availability Statement:** All relevant data are within the paper and its Supporting Information files.

## Abstract

### Background

Abnormal fetal growth can be associated with factors during pregnancy and at postpartum.

### Objective

In this study, we aimed to assess the incidence, risk factors, and feto-maternal outcomes associated with small-for-gestational age (SGA) and large-for-gestational age (LGA) infants.

### Methods

We performed a population-based retrospective study on 14,641 singleton live births registered in the PEARL-Peristat Study between April 2017 and March 2018 in Qatar. We estimated the incidence and examined the risk factors and outcomes using univariate and multivariate analysis.

### Results

SGA and LGA incidence rates were 6.0% and 15.6%, respectively. In-hospital mortality among SGA and LGA infants was 2.5% and 0.3%, respectively, while for NICU admission

**Funding:** The PEARL-Peristat study was funded by Qatar National Research Fund (Grant no NPRP 6-238-3-059) and was sponsored by the Medical Research Centre, Hamad Medical Corporation. The funders had no role in study design, data collection and analysis, decision to publish, or preparation of the manuscript.

**Competing interests:** The authors have declared that no competing interests exist.

or death in labor room and operation theatre was 28.9% and 14.9% respectively. Preterm babies were more likely to be born SGA (aRR, 2.31; 95% CI, 1.45–3.57) but male infants (aRR, 0.57; 95% CI, 0.4–0.81), those born to parous (aRR 0.66; 95% CI, 0.45–0.93), or overweight (aRR, 0.64; 95% CI, 0.42–0.97) mothers were less likely to be born SGA. On the other hand, males (aRR, 1.82; 95% CI, 1.49–2.19), infants born to parous mothers (aRR 2.16; 95% CI, 1.63–2.82), or to mothers with gestational diabetes mellitus (aRR 1.36; 95% CI, 1.11–1.66), or pre-gestational diabetes mellitus (aRR 2.58; 95% CI, 1.8–3.47) were significantly more likely to be LGA. SGA infants were at high risk of in-hospital mortality (aRR, 226.56; 95% CI, 3.47–318.22), neonatal intensive care unit admission or death in labor room or operation theatre (aRR, 2.14 (1.36–3.22).

## Conclusion

Monitoring should be coordinated to alleviate the risks of inappropriate fetal growth and the associated adverse consequences.

## 1. Introduction

Gestational age and birth weight are two crucial factors for assessing the fetal growth. Birth weight is a strong determinant of a newborn infant's survival rate [1, 2]. An appropriate birth weight at gestational age (AGA) is critical when assessing the typical development of a newborn infant. Inappropriate gestational age classification ranges from small-for-gestational age (SGA), referring to birth weight below the 10th percentile, and large-for-gestational age (LGA), referring to birth weight above the 90th percentile [3].

There is a substantial disparity in the prevalence of SGA babies (4.6–15.3%) across Europe [4] and LGA babies (5–20%) in developed countries [5]. These varieties are more apparent in developing countries. According to global estimates, in 2010, 27% of all live births were found to be SGA (over 32 million) in low- and middle-income countries [6], with an SGA prevalence as high as 41.5% in Pakistan and as low as 5.3% in China [7]. There is also a huge deviation in the prevalence of macrosomia (birthweight ≥4000 g) in developing countries, with figures as low as 0.5% in India and as high as 14.9% in Algeria [8]. The variability in the rates of prevalence of SGA and LGA infants is mainly due to socio-environmental factors, population differences, as well as wide variations in the standards applied for assessment in different studies [8, 9].

Size for gestational age is considered as a measure of fetal growth, with SGA regarded an indication of fetal growth restriction and LGA as an indication of rapid fetal development [10, 11]. Risk factors that have been linked to SGA include pre-pregnancy weight, previous history of SGA, smoking, and cardiovascular-associated diseases [12–17]. On the other hand, maternal obesity, diabetes, multipara was found to be linked to higher rates of LGA [12, 15, 17–20].

Babies born SGA or LGA are at high risk of developing increased long-term health complications during the antepartum, intrapartum, and postpartum periods. SGA infants have been shown to develop health complications including birth asphyxia, hypothermia and abnormal neurological development, and are at high risk of mortality [21–28], whereas LGA infants have been shown to develop postpartum hemorrhage and birth injuries [5, 18, 29]. Thus, these newborns often need specialized care to avoid and manage the complications.

Several studies have investigated the risk factors and outcomes associated with birth weight and gestational age separately. However, the concept of defining birth weight in the context of

gestational age, referred to as 'birthweight percentiles' has been understudied specifically in the Middle East [30]. In addition, most studies to date have focused on low birth weight, and only few reports have described the link between increased birth weight and high mortality risks [31–33] or death in the Neonatal Intensive Care Unit (NICU). While SGA is generally known to be associated with several neonatal outcomes [1–8], LGA is understudied, and comparisons between both groups with AGA in the context of risk factors and outcomes are lacking.

SGA or LGA have traditionally been defined using standards that were based on the weight distribution of infants born in a particular population, rather than describing physiological or healthy growth [34]. In fact, most studies have advocated the continued use of local or customized charts [35, 36]; however, these local charts are only relevant to the population and time from which they were derived and hence make comparison between populations and studies impossible. Recently, The International Fetal and Newborn Growth Consortium for the Twenty-First Century (INTERGROWTH-21st) has described a multinational standard for newborn weight. This research revealed that when women who are not subjected to societal, dietary, medical, or other restrictions on fetal growth, the growth of infants all over the globe is surprisingly comparable [34]. Thus, the INTERGROWTH-21st birth weight standard offers a reliable multinational tool for estimating fetal weight percentiles.

In the present study we aimed to assess the incidence, risk factors and feto-maternal outcomes associated with SGA and LGA births via a population-based retrospective data analysis of singleton live births data retrieved from the PEARL-Peristat Study between April 2017 and March 2018 in Qatar. We examined several demographic and medical confounders to assess the risk for SGA and LGA, while investigating how these confounders are associated with low Apgar score, NICU admission, and mortality. In addition, we explored the relationship between inappropriate birth weight for gestational age and preterm birth, taking into account late preterm and early terms which are rarely investigated in the literature.

## 2. Methods

### 2.1. Study design

This was a 12-month retrospective population-based study conducted using registry data from the PEARL-Peristat Study, Qatar. This population-based registry was designed using routinely collected hospital data for parturient women and their offspring. The study was approved by the Hamad Medical Corporation (HMC) Institutional Review Board (IRB), with a waiver of consent.

We included singleton live births at 24+0 weeks gestation and above, whose mothers delivered between April 2017 and March 2018 at the Women's Wellness and Research Centre (WWRC) in HMC. HMC is the main national hospital, and the main provider of secondary and tertiary healthcare in Qatar. It is also one of the leading hospital providers in the Middle East. HMC consists of four regional hospitals that are widely distributed in different geographical areas of Qatar (Al-Wakra, Al-Khor, Cuban and Women's Wellness Research Centre hospitals). These hospitals account for the majority of births in the country. In addition, premature babies and those who are admitted to NICU come to these hospitals only. Stillbirths were excluded. A total of 14,641 singleton births were examined.

**2.1.1. Neonatal factors.** We used the FETALGPSXL tool [37, 38] which takes into account gestational age (days), fetal weight (grams), gender, and maternal ethnic/race group to calculate fetal weight percentiles for births occurring prior to 280 days. This tool provides a simple spreadsheet-based estimated fetal weight percentile calculator and corresponding R software package encompassing 6 fetal growth standards, among which we chose the Intergrowth 21st

standard to calculate the percentiles [38]. Accordingly, newborns were categorized into three groups: SGA (defined as birth weight for gestational age below the 10th percentile), AGA (defined as birth weight for gestational age between the 10th and the 90th percentile; reference group), and LGA (defined as birthweight for gestational age above the 90th percentile) [39].

Gestational age (GA) was based on mother's last menstrual period (LMP), early ultrasound scan (USS) and Ballard scoring [40]. GA was classified in accordance with established international definitions [41]; into preterm (less than 37 weeks' gestation) and term (at 37 weeks' gestation and above). For further investigation, GA was further categorized into; extreme to very preterm: < 32 weeks, moderate preterm: 32 to < 34 weeks, late preterm: 34 to < 37 weeks, early term: 37 to < 39, and full term: 39 to < 42). Baby gender was categorized into male, female, and ambiguous. Immediate birth status included an Apgar (Appearance, Pulse, Grimace, Activity, and Respiration) score < 7 at 1 minute, and at 5 minutes. Baby outcome was categorized into discharged alive or in-hospital mortality, while baby disposition was categorized into postnatal ward and NICU or death in Labour Room/ Operation Theatre (LR/OT).

**2.1.2. Maternal factors.** Maternal age at delivery was grouped into young adults (20–34 years), adolescents (<20 years), and advanced maternal age (> 35 years). Nationality was grouped into Qatari, other Arabs and other nationalities based on the UNESCO list of Arab countries. Consanguinity was coded as yes (the mother and the father are related to each other in any level of relatedness) or no. Educational level was classified into elementary and below, secondary school or high school, college/university or above. Employment status was categorised into employed or unemployed. Smoking status was coded as yes or no, where the mother was asked whether she is a smoker or not.

Women were categorized according to their glycemic status into diabetic and non-diabetic, and further categorized into pregestational diabetics (PGDM), gestational diabetics (GDM) and non-diabetics (no data on Type 1 or 2 were recorded). All pregnant women were screened at the first antenatal care visit using fasting blood glucose and HBA1c- to rule out pre-existing diabetes. Then, 75 grams oral glucose tolerance test (OGTT) was performed between 24–32 weeks' gestation in low-risk patients and between 16–20 weeks' gestation in high-risk patients. GDM was diagnosed according to the modified International Association of Diabetes and Pregnancy Study Groups (IADPSG) criteria [42], when one or more of the following glucose levels were elevated: fasting plasma glucose level $\geq$5.1 mmol/L, 1 h plasma glucose level $\geq$10.0 mmol/L, and 2 h plasma glucose level $\geq$8.5 mmol/L [42].

Chronic hypertension was coded as yes or no. In addition, for Body Mass Index (BMI) we used pre-pregnancy height and weight and in case they are not available, the early pregnancy (gestational age < = 12) weight and height were used. These measures were taken by the health practitioner (doctor or nurse). Accordingly, mothers' BMI was categorized into four groups: normal (18.5 to 24.9), underweight (< 18.5), overweight, (25.0 to 29.9) and obese ($\geq$ 30 kg/ m$^2$) following NHLBI/WHO guidelines [43, 44].

Parity was classified into nulliparous or parity $\geq$1. A history of any preterm birth (spontaneous or medically indicated) was coded as yes or no. Pregnancy mode was defined as spontaneous or assisted (including ovulation induction, invitro fertilisation, intracytoplasmic sperm injection, intra uterine insemination, and others). Delivery mode was categorized into vaginal and caesarean.

## 2.2. Statistical analysis

Statistical analysis was conducted using IBM SPSS 26 software (SPSS Chicago IL, USA). All categorical and binary variables were expressed as numbers and percentages. The overall incidence of SGA and LGA, risk factors, and outcomes were analyzed using Chi Square analysis.

Firstly, logistic regression analysis was performed for risk factors/confounders (demographic and medical factors) and mediators (prematurity and gender) of appropriateness of fetal growth for the GA groups (SGA/LGA vs. AGA). In step one univariate analysis was performed, and the associations were quantified. The statistical significance was set at p<0.05. In step two, multiple logistic regression was performed using all the significant variables (P<0.05) from the univariate analysis as confounders (demographic and medical factors), along with the mediators (prematurity and gender) to investigate associations with SGA and LGA groups.

Secondly, logistic regression was performed to investigate the outcomes of SGA and LGA including Apgar score, NICU/death in LR/OT, and in-hospital mortality. Multiple logistic regression was performed, including all significant confounders (prematurity and gender) from the univariate analysis to investigate the association of SGA/LGA with Apgar score, NICU/death in LR/OT, and in-hospital mortality as outcomes.

We then applied the formula described by Zhang and Yu [45], to compute the relative risk (RR) from the odds ratio (OR) for all logistic regression analyses. Crude and adjusted RRs and their 95% CIs were recorded, with a statistical significance set at p<0.05.

Furthermore, we calculated the population attributable fraction (PAF) % among the different risk factors to determine what percentage of SGA and LGA births might have been prevented if the risk factors had been avoided. For calculating the crude PAFs (cPAFs), we utilized the formula $PAF = P_e (RR_e - 1)/[1 + P_e (RR_e - 1)]$ [46–48], where $P_e$ is the percentage of people in the population who were exposed to the risk factor and $RR_e$ is the crude relative risk in the exposed vs. the unexposed group. For the adjusted PAFs (aPAFs), we used the formula $P_d [(aRR-1)/aRR]$, in which $P_d$ is the prevalence of exposure among those who were born SGA or LGA, and $_aRR$ is the adjusted relative risk in the exposed vs. unexposed group [49–51].

Kaplan-Meier curves were constructed to assess differences in medians, among the three groups (AGA, SGA and LGA) for the outcomes (Apgar score, NICU/death in LR/OT and in-hospital mortality) during the course of 24–40 weeks of gestation. A log rank (Mantel Cox) test was used to assess this difference, with a two-tailed P-value <0.05 regarded as statistically significant.

## 3. Results

### 3.1. Characteristics of the study population

A total of 14,641 singleton live births registered in the PEARL database from April 2017 to March 2018 were examined. Of these, 32.45% were overweight mothers and 32.34% were obese. In addition, 31.68% of the mothers had total DM (29.09% GDM and 2.6% PGDM). The maternal characteristics and distribution of the overall study population according to fatal growth are presented in Table 1 and S1 Table. SGA and LGA incidence rates were 6.0% and 15.6%, respectively. In-hospital mortality was 2.5% among SGA infants and 0.3% among LGA infants, while NICU admission or death in LR or OT were 28.9% and 14.9% respectively (Table 1).

There was a significant difference among the groups in the distribution of SGA and LGA in term of gestational age, maternal age, parity, nationality, education, diabetes status, chronic hypertension, early- or pre-pregnancy BMI, baby gender, chromosomal/congenital abnormalities, employment status, delivery mode, Apgar <7 at 1 min, Apgar <7 at 5 mins, baby outcome, and baby disposition (P<0.05). SGA was more likely to occur amongst female preterm babies who were born to adolescent underweight mothers from other nationalities, Qataris, with chronic hypertension and with more chromosomal/congenital abnormalities, with low

**Table 1. Characteristics of the study population.**

| | AGA (n = 11,477) | SGA (n = 882) | LGA (n = 2,282) | |
|---|---|---|---|---|
| | n (%) | n (%) | n (%) | *p* value |
| Gestational Age | | | | **0.000** |
| Preterm | 898 (7.8) | 181 (20.5) | 309 (13.5) | |
| Term | 10579 (92.2) | 701 (79.5) | 1973 (86.5) | |
| Maternal age | | | | **0.000** |
| Young adults (20–34 years) | 8971 (78.2) | 704 (79.8) | 1678 (73.5) | |
| Adolescents (<20 years) | 252 (2.2) | 35 (4) | 29 (1.3) | |
| Advanced maternal age (≥35 years) | 2254 (19.6) | 143 (16.2) | 575 (25.2) | |
| Parity | | | | **0.000** |
| Nulliparous | 3186 (27.8) | 379 (43) | 341 (14.9) | |
| Parity ≥1 | 8291 (72.2) | 503 (57) | 1941 (85.1) | |
| Pregnancy mode | | | | 0.211 |
| Spontaneous | 11105 (97.3) | 851 (97.5) | 2221 (97.9) | |
| Assisted | 310 (2.7) | 22 (2.5) | 47 (2.1) | |
| Nationality | | | | **0.000** |
| Qatari | 3620 (31.5) | 308 (34.9) | 616 (27) | |
| Other Arabs | 4450 (38.8) | 261 (29.6) | 1072 (47) | |
| Other Nationalities | 3404 (29.7) | 313 (35.5) | 593 (26) | |
| Consanguinity | | | | 0.655 |
| No | 3064 (66.4) | 204 (64.4) | 584 (67.2) | |
| Yes | 1547 (33.6) | 113 (35.6) | 285 (32.8) | |
| Education | | | | **0.005** |
| Elementary and below | 425 (8.6) | 27 (7.8) | 102 (11) | |
| Secondary/Highschool | 1523 (30.8) | 127 (36.8) | 254 (27.3) | |
| University or above | 2992 (60.6) | 191 (55.4) | 574 (61.7) | |
| Diabetes Status | | | | **0.000** |
| No DM | 8000 (69.7) | 654 (74.1) | 1348 (59.1) | |
| GDM | 3235 (28.2) | 214 (24.3) | 810 (35.5) | |
| PGDM | 242 (2.1) | 14 (1.6) | 124 (5.4) | |
| Chronic Hypertension | | | | **0.000** |
| No | 11335 (98.8) | 857 (97.2) | 2244 (98.3) | |
| Yes | 142 (1.2) | 25 (2.8) | 38 (1.7) | |
| Pre- or early-pregnancy BMI | | | | **0.000** |
| Normal | 1494 (33.7) | 145 (43) | 196 (22.2) | |
| Underweight | 128 (2.9) | 21 (6.2) | 5 (0.6) | |
| Overweight | 1445 (32.6) | 91 (27) | 297 (33.7) | |
| Obese | 1364 (30.8) | 80 (23.7) | 383 (43.5) | |
| Baby gender | | | | **0.000** |
| Male | 5705 (49.7) | 348 (39.5) | 1443 (63.2) | |
| Female | 5769 (50.3) | 534 (60.5) | 839 (36.8) | |
| Ambiguous | 2 (0) | 0 (0) | 0 (0) | |
| Chromosomal/Congenital abnormalities | | | | **0.000** |
| No | 11319 (98.6) | 830 (94.1) | 2244 (98.3) | |
| Yes | 158 (1.4) | 52 (5.9) | 38 (1.7) | |
| Smoking | | | | 0.589 |
| No | 8767 (99.1) | 664 (98.8) | 1708 (99.2) | |
| Yes | 77 (0.9) | 8 (1.2) | 13 (0.8) | |

(*Continued*)

**Table 1.** (Continued)

| | AGA (n = 11,477) | SGA (n = 882) | LGA (n = 2,282) | |
|---|---|---|---|---|
| | n (%) | n (%) | n (%) | p value |
| Preterm history | | | | 0.220 |
| No | 10701 (93.2) | 814 (92.3) | 2108 (92.4) | |
| Yes | 776 (6.8) | 68 (7.7) | 174 (7.6) | |
| Employment status | | | | **0.004** |
| Employed | 4648 (99.1) | 319 (98.8) | 874 (97.8) | |
| Unemployed | 44 (0.9) | 4 (1.2) | 20 (2.2) | |
| Delivery mode | | | | **0.000** |
| Vaginal | 8193 (71.4) | 554 (62.8) | 1323 (58) | |
| Caesarean | 3284 (28.6) | 328 (37.2) | 959 (42) | |
| Apgar <7 at 1 min | | | | **0.000** |
| No | 11241 (98.2) | 807 (92.2) | 2215 (97.3) | |
| Yes | 208 (1.8) | 68 (7.8) | 62 (2.7) | |
| Apgar <7 at 5 mins | | | | **0.000** |
| No | 11430 (99.8) | 867 (99) | 2267 (99.6) | |
| Yes | 26 (0.2) | 9 (1) | 9 (0.4) | |
| Baby disposition | | | | **0.000** |
| Postnatal ward | 10346 (90.2) | 627 (71.1) | 1941 (85.1) | |
| NICU or died in LR/OT | 1130 (9.8) | 255 (28.9) | 341 (14.9) | |
| Baby outcome | | | | **0.000** |
| Discharged alive | 11441 (99.7) | 860 (97.5) | 2276 (99.7) | |
| Died in hospital | 36 (0.3) | 22 (2.5) | 6 (0.3) | |

Abbreviations: DM, diabetes mellitus; GDM, gestational diabetes mellitus; PGDM, pre-gestational diabetes mellitus; AGA, appropriate for gestational age; SGA, small for gestational age; LGA, large for gestational age; Apgar, Appearance, Pulse, Grimace, Activity, and Respiration; NICU, neonatal intensive care unit; LR, labour room; OT, operation theatre.

**Bold** values denote statistical significance at the p<0.05 level.

Apgar score <7 at 1 minute and 5 minutes (p<0.05). On the other hand, LGA was more likely to occur amongst babies with advanced age mothers, parity ≥1, from other Arab origin, with GDM and PGDM and overweight and obese mothers (p<0.05).

## 3.2. Risk factors associated with inappropriate weight for gestational age

Univariate analysis for SGA as an outcome revealed that preterm birth (cRR, 2.7; 95% CI, 2.32–3.14) and male baby gender (cRR, 0.68; 95% CI, 0.6–0.77) were significantly related to SGA. In addition, SGA was more likely to occur amongst babies of adolescent mothers (cRR, 1.68; 95% CI, 1.22–2.3), with a secondary/high school level of education (cRR, 1.28 (1.03–1.59), chronic hypertension (cRR, 2.13; 95% CI, 1.48–3.07), underweight (cRR, 1.59; 95% CI, 1.04–2.44), or with chromosomal/congenital abnormalities (cRR, 3.55; 95% CI, 2.75–4.58). On the other hand, SGA was less likely to occur amongst mothers with advanced age (0.82 (0.69–0.98), from other Arabs origin (cRR, 0.66; 95% CI, 0.56–0.77), parity ≥1 (cRR, 0.54; 95% CI, 0.47–0.61), GDM (cRR, 0.82; 95% CI, 0.71–0.95), overweight (cRR, 0.67; 95% CI, 0.52–0.86), and obese (cRR, 0.63; 95% CI, 0.48–0.82). In the multivariate analysis, preterm birth (aRR, 2.31; 95% CI, 1.45–3.57) and baby gender (aRR, 0.57; 95% CI, 0.4–0.81) remained significant mediators. In addition, the confounding variables; parity (aRR, 0.66; 95% CI, 0.45–0.93), and overweight mothers (aRR, 0.64; 95% CI, 0.42–0.97) remained significant. The rest of the factors became non-significant in the adjusted model (Table 2).

**Table 2. Risk factors associated with SGA and LGA.**

| Risk factors | SGA[a] | | | | | | LGA[b] | | | | | |
|---|---|---|---|---|---|---|---|---|---|---|---|---|
| | (n = 882) | | | | | | (n = 2282) | | | | | |
| | cRR | p value | cPAF (%) | aRR | p value | aPAF (%) | cRR | p value | cPAF (%) | aRR | p value | aPAF (%) |
| Gestational Age* | | | | | | | | | | | | |
| Preterm | 2.7 (2.32–3.14) | **0.000** | 12.9 | 2.31 (1.45–3.57) | **0.001** | 11.641 | 1.63 (1.47–1.81) | **0.000** | 5.2 | 1.5 (1.07–2.02) | **0.019** | 4.5 |
| Term | Ref | | 0.0 | Ref | | | Ref | | 0.0 | Ref | | |
| Maternal age | | | | | | | | | | | | |
| young adults (20–34 years) | Ref | | 0.0 | Ref | | | Ref | | 0.0 | Ref | | |
| adolescents (<20 years) | 1.68 (1.22–2.3) | **0.002** | 1.9 | 1.63 (0.72–3.44) | 0.240 | 1.8378 | 0.65 (0.46–0.93) | **0.013** | -0.9 | 0.62 (0.15–2.04) | 0.471 | -1.0 |
| Advanced maternal age (≥35 years) | 0.82 (0.69–0.98) | **0.025** | -3.7 | 0.83 (0.5–1.35) | 0.465 | -3.436 | 1.29 (1.18–1.4) | **0.000** | 5.7 | 0.96 (0.74–1.21) | 0.707 | -1.1 |
| Parity | | | | | | | | | | | | |
| Nulliparous | Ref | | 0.0 | Ref | | | Ref | | 0.0 | Ref | | |
| Parity ≥1 | 0.54 (0.47–0.61) | **0.000** | -48.6 | 0.66 (0.45–0.93) | **0.018** | -29.94 | 1.96 (1.76–2.19) | **0.000** | 41.7 | 2.16 (1.63–2.82) | **0.000** | 45.7 |
| Pregnancy Mode | | | | | | | | | | | | |
| Spontaneous | Ref | | 0.0 | | | | Ref | | 0.0 | | | |
| Assisted | 0.93 (0.62–1.4) | 0.731 | -0.2 | | | | 0.79 (0.6–1.03) | 0.079 | -0.6 | | | |
| Nationality | | | | | | | | | | | | |
| Qatari | 0.93 (0.8–1.08) | 0.354 | -3.7 | 1.41 (0.94–2.07) | 0.104 | 14.314 | 0.98 (0.88–1.09) | 0.706 | -1.0 | 1.11 (0.82–1.47) | 0.500 | 5.0 |
| Other Arabs | 0.66 (0.56–0.77) | **0.000** | -23.4 | 0.84 (0.54–1.28) | 0.423 | -8.531 | 1.31 (1.19–1.43) | **0.000** | 15.2 | 1.53 (1.2–1.91) | **0.001** | 22.2 |
| Other Nationalities | Ref | | 0.0 | Ref | | | Ref | | 0.0 | Ref | | |
| Consanguinity | | | | | | | | | | | | |
| No | Ref | | 0.0 | | | | Ref | | 0.0 | | | |
| Yes | 1.09 (0.87–1.36) | 0.445 | 2.9 | | | | 0.97 (0.85–1.11) | 0.666 | -1.0 | | | |
| Education | | | | | | | | | | | | |
| Elementary and below | 1 (0.67–1.47) | 0.982 | 0.0 | 1.01 (0.53–1.89) | 0.976 | 0.1153 | 1.2 (0.99–1.45) | 0.060 | 2.5 | | | |
| Secondary/Highschool | 1.28 (1.03–1.59) | **0.024** | 8.7 | 1.38 (0.95–1.96) | 0.085 | 10.916 | 0.89 (0.77–1.02) | 0.086 | -3.8 | | | |
| University or above | Ref | | 0.0 | Ref | | | Ref | | 0.0 | | | |
| Diabetes Status | | | | | | | | | | | | |
| No DM | Ref | | 0.0 | Ref | | | Ref | | 0.0 | Ref | | |
| GDM | 0.82 (0.71–0.95) | **0.009** | -5.4 | 0.89 (0.61–1.29) | 0.551 | -3.106 | 1.39 (1.28–1.5) | **0.000** | 10.5 | 1.36 (1.11–1.66) | **0.004** | 10.0 |
| PGDM | 0.72 (0.43–1.21) | 0.211 | -0.8 | 0.42 (0.1–1.72) | 0.237 | -2.905 | 2.35 (2.02–2.73) | **0.000** | 4.8 | 2.58 (1.8–3.47) | **0.000** | 5.2 |
| Chronic Hypertension | | | | | | | | | | | | |
| No | Ref | | 0.0 | Ref | | | Ref | | 0.0 | | | |
| Yes | 2.13 (1.48–3.07) | **0.000** | 1.5 | 2.58 (0.78–6.53) | 0.116 | 1.7364 | 1.28 (0.96–1.7) | 0.100 | 0.4 | | | |
| Pre-or early-pregnancy BMI | | | | | | | | | | | | |
| Normal | Ref | | 0.0 | Ref | | | Ref | | 0.0 | Ref | | |
| Underweight | 1.59 (1.04–2.44) | **0.035** | 4.7 | 1.54 (0.78–2.84) | 0.212 | 4.4155 | 0.32 (0.14–0.77) | **0.005** | -5.3 | 0.56 (0.18–1.62) | 0.300 | -2.0 |

(*Continued*)

**Table 2.** (Continued)

| Risk factors | SGA[a] | | | | | | LGA[b] | | | | | |
| --- | --- | --- | --- | --- | --- | --- | --- | --- | --- | --- | --- | --- |
| | (n = 882) | | | | | | (n = 2282) | | | | | |
| | cRR | p value | cPAF (%) | aRR | p value | aPAF (%) | cRR | p value | cPAF (%) | aRR | p value | aPAF (%) |
| Overweight | 0.67 (0.52–0.86) | **0.002** | -19.0 | 0.64 (0.42–0.97) | **0.037** | -21.55 | 1.47 (1.24–1.74) | **0.000** | 19.3 | 1.12 (0.86–1.46) | 0.410 | 6.5 |
| Obese | 0.63 (0.48–0.82) | **0.000** | -20.9 | 0.65 (0.41–1.02) | 0.060 | -19.04 | 1.89 (1.61–2.22) | **0.000** | 31.1 | 1.15 (0.87–1.5) | 0.329 | 8.5 |
| Baby gender | | | | | | | | | | | | |
| Male | 0.68 (0.6–0.77) | **0.000** | -18.6 | 0.57 (0.4–0.81) | **0.002** | -29.54 | 1.59 (1.47–1.72) | **0.000** | 23.5 | 1.82 (1.49–2.19) | **0.000** | 28.5 |
| Female | Ref | | 0.0 | Ref | | | Ref | | 0.0 | Ref | | |
| Chromosomal/Congenital abnormalities | | | | | | | | | | | | |
| No | Ref | | 0.0 | Ref | | | Ref | | 0.0 | | | |
| Yes | 3.55 (2.75–4.58) | **0.000** | 4.2 | 2.03 (0.79–4.56) | 0.137 | 2.9858 | 1.18 (0.88–1.58) | 0.290 | 0.3 | | | |
| Smoking | | | | | | | | | | | | |
| No | Ref | | 0.0 | | | | Ref | | 0.0 | | | |
| Yes | 1.34 (0.69–2.6) | 0.396 | 0.3 | | | | 0.89 (0.53–1.47) | 0.634 | -0.1 | | | |
| Preterm history | | | | | | | | | | | | |
| No | Ref | | 0.0 | | | | Ref | | 0.0 | | | |
| Yes | 1.14 (0.9–1.45) | 0.282 | 0.9 | | | | 1.11 (0.97–1.28) | 0.137 | 0.8 | | | |
| Employment status | | | | | | | | | | | | |
| Employed | Ref | | 0.0 | | | | Ref | | 0.0 | Ref | | |
| Unemployed | 1.3 (0.5–3.34) | 0.591 | 0.3 | | | | 1.97 (1.37–2.85) | **0.001** | 1.1 | 2.37 (1.2–3.81) | **0.015** | 1.3 |

Abbreviations; cRR, crude risk ratio; aRR, adjusted risk ratio; cPAF, crude population attributable fraction; aPAF, adjusted population attributagle fraction; CI, confidence interval; Ref, referent; DM, diabetes mellitus; GDM, gestational diabetes mellitus; PGDM, pre-gestational diabetes mellitus; AGA, appropriate for gestational age; SGA, small for gestational age; LGA, large for gestational age; Apgar, Appearance, Pulse, Grimace, Activity, and Respiration; NICU, neonatal intensive care unit; LR, labour room; OT, operation theatre.

For the cPAFs, we utilized the formula $PAF = P_e (RRe - 1)/[1 + Pe (RRe - 1)]$ [43–45] where $P_e$ is the percentage of people in the population who were exposed to the risk factor and $RR_e$ is the crude relative risk in the exposed vs. the unexposed group. For the adjusted PAFs (aPAFs), we used the formula $P_d ((aRR—1)/aRR)$, in which $P_d$ is the prevalence of exposure among those who were born SGA or LGA, and aRR is the adjusted relative risk in the exposed vs. unexposed group [49–51]. The full results are provided in S2 Table.

[a] adjusted for the risk factors associated with SGA that were significant in the univariate analysis, with p-values <0.05: Gestational age, maternal age, parity, nationality, education, diabetes status, chronic hypertension, early- or pre-pregnancy BMI, baby gender, and any chromosomal or congenital abnormalities.

[b] adjusted for the risk factors associated with LGA that were significant in the univariate analysis, with p-values <0.05: Gestational age, maternal age, parity, nationality, diabetes status, early- or pre-pregnancy BMI, baby gender, and employment status.

*The analysis presented here was conducted using the gestational age variable categorized into two groups. The full analysis was reconducted with gestational age categorized into five groups, and presented in S3 Table.

**Bold** values denote statistical significance at the p<0.05 level.

Univariate analysis for LGA as an outcome revealed that preterm birth (cRR, 1.63; 95% CI, 1.47–1.81), and male baby gender (cRR, 1.59; 95% CI, 1.47–1.72) were significantly related to LGA. In addition, LGA was more likely to occur amongst babies of mothers with advanced age (cRR, 1.29; 95% CI, 1.18–1.4), with parity ≥1 (cRR, 1.96; 95% CI, 1.76–2.19), from other Arabs origin (cRR, 1.31; 95% CI, 1.19–1.43), with GDM (cRR, 1.39; 95% CI, 1.28–1.5) and PGDM

(cRR, 2.35; 95% CI, 2.02–2.73), who are overweight (cRR, 1.47; 95% CI, 1.24–1.74), obese (cRR, 1.89; 95% CI, 1.61–2.22), and unemployed (cRR, 1.97; 95% CI, 1.37–2.85). While LGA was less likely to happen with babies of adolescent mothers (cRR, 0.65 (0.46–0.93), and underweight (cRR, 0.32; 95% CI, 0.14–0.77). In the multivariate analysis, the mediators; preterm birth (aRR, 1.5; 95% CI, 1.07–2.02) and male baby gender (aRR, 1.82; 95% CI, 1.49–2.19), in addition to the confounding variables; parity (aRR, 2.16; 95% CI, 1.63–2.82), other Arabs (aRR, 1.53; 95% CI, 1.2–1.91), GDM (aRR, 1.36; 95% CI, 1.11–1.66), PGDM (aRR, 2.58; 95% CI, 1.8–3.47) were found to be significantly associated with LGA. The rest of the confounders became non-significant in the adjusted model (Table 2).

The highest aPAF among SGA births was observed for preterm birth, with an aPAF of 11.6%, indicating that almost 12% of SGA cases could have been prevented if mothers had not delivered preterm (Table 2 and S2 Table), whereas LGA preterm infants showed 4.5% for aPAF, indicating that only 5% of LGA cases could have been prevented if mothers had not delivered preterm (Table 2 and S2 Table). Among LGAs, the highest aPAF was 45.7% for Parity ≥1, indicating that almost half of the LGA cases could have been prevented if mothers were not parous, whereas SGA infants showed a negative aPAF of -29.9%, indicating that Parity ≥1 is a protective factor for SGA birth.

Univariate analysis with the gestational age categorized into five categories revealed that extreme to very preterm (cRR, 3.9; 95% CI, 2.95–5.15), moderate preterm (cRR, 3.46; 95% CI, 2.47–4.85), and late preterm (cRR, 2.16; 95% CI, 1.78–2.62) were significantly associated to a higher risk of SGA (S3 Table). Following adjustment for the confounding factors in the multivariate analysis, only extreme to very preterm (aRR, 3.07; 95% CI, 1.01–7.22), and late preterm birth (aRR, 1.92; 95% CI, 1.1–3.22) remained significant mediators for SGA (S3 Table). For LGA, we found that all five groups were significantly associated with LGA in the univariate model (S3 Table). However, following adjustment for the confounding factors, only late preterm (aRR, 1.68; 95% CI, 1.14–2.39) and early term (aRR, 1.4; 95% CI, 1.13–1.71) were found to be significant mediators for LGA birth (S3 Table).

## 3.3. Adverse outcomes associated with inappropriate weight for gestational age

Univariate logistic regression analysis revealed that SGA in comparison to AGA, was significantly associated with low Apgar <7 at 1 min (cRR, 4.28; 95% CI, 3.28–5.58), low Apgar <7 at 5 mins (cRR, 4.53; 95% CI, 2.13–9.63), NICU/death in LR/OT (cRR, 2.94; 95% CI, 2.61–3.3), and in-hospital mortality (cRR, 7.95; 95% CI, 4.7–13.46) (Table 3 and S4 Table). After adjustment, SGA was significantly associated with NICU/death in LR/OT (aRR, 2.14; 95% CI, 1.36–3.22) and in-hospital mortality (aRR, 226.56; 95% CI, 3.47–318.22) (Table 3). However, the relationship of SGA with low Apgar <7 at 1 min and 5 minutes became non-significant after adjustment (Table 3 and S4 Table).

Univariate logistic regression analysis revealed that LGA, compared to AGA, was significantly associated with low Apgar <7 at 1 min (cRR, 1.50; 95% CI, 1.13–1.98) and NICU/death in LR/OT (cRR, 1.52; 95% CI, 1.36–1.7) (S4 Table). However, after adjustment the association of LGA with low Apgar <7 at 1 min, and NICU/death in LR/OT became non-significant (Table 3).

Univariate analyses with the gestational age categorized into five categories revealed that extreme to very preterm (cRR, 32.61; 95% CI, 26.11–40.74), moderate preterm (cRR, 8.85; 95% CI, 5.68–13.79), late preterm (cRR, 3.85; 95% CI, 2.83–5.22) were significantly associated with a higher risk of low Apgar score <7 at 1 minute, while early term was significantly associated with lower risk of low Apgar score <7 at 1 minute (cRR, 0.73; 95% CI, 0.53–1.01). Following

**Table 3. Outcomes associated with SGA and LGA.**

| | Apgar <7 at 1 min | | | | NICU/death in LR/OT | | | | In–hospital mortality | | | |
|---|---|---|---|---|---|---|---|---|---|---|---|---|
| | cRR (95% CI) | p value | aRR (95% CI) | p value | cRR (95% CI) | p value | aRR (95% CI) | p value | cRR (95% CI) | p value | aRR (95% CI) | p value |
| Table 3–A: SGA[a] vs. AGA | | | | | | | | | | | | |
| SGA | 4.28 (3.28–5.58) | **0.000** | 1.59 (0.56–4.33) | 0.372 | 2.94 (2.61–3.3) | **0.000** | 2.14 (1.36–3.22) | **0.002** | 7.95 (4.7–13.46) | **0.000** | 226.56 (3.47–318.22) | **0.016** |
| AGA | Ref | | Ref | | Ref | | Ref | | Ref | | Ref | |
| Gestational Age* | | | | | | | | | | | | |
| Preterm | 9.79 (8.03–11.95) | **0.000** | 8.41 (4.23–15.81) | **0.000** | 7.05 (6.54–7.6) | **0.000** | 7.07 (5.7–8.4) | **0.000** | 19.94 (12.08–32.91) | **0.000** | 536.79 (5.66–629.22) | **0.013** |
| Term | Ref | | Ref | | Ref | | Ref | | Ref | | Ref | |
| Maternal Age | | | | | | | | | | | | |
| Young adults (20–34 years) | Ref | | Ref | | Ref | | Ref | | Ref | | Ref | |
| Adolescents (<20 years) | 1.44 (0.79–2.6) | 0.230 | 0.94 (0.11–7.17) | 0.954 | 1.01 (0.75–1.36) | 0.956 | 0.61 (0.17–2.01) | 0.442 | 0.78 (0.11–5.67) | 0.809 | N/A | |
| Advanced maternal age (≥35 years) | 1.17 (0.92–1.5) | 0.210 | 1.04 (0.4–2.68) | 0.936 | 1.23 (1.11–1.36) | **0.000** | 1.74 (1.2–2.43) | **0.004** | 1.82 (1.09–3.05) | **0.021** | 18.22 (0.58–193.76) | 0.098 |
| Parity | | | | | | | | | | | | |
| Nulliparous | Ref | | Ref | | Ref | | Ref | | Ref | | Ref | |
| Parity ≥1 | 0.46 (0.38–0.57) | **0.000** | 0.55 (0.25–1.19) | 0.132 | 0.64 (0.59–0.7) | **0.000** | 0.61 (0.42–0.88) | **0.007** | 0.88 (0.52–1.48) | 0.621 | 3.75 (0.09–94.59) | 0.490 |
| Nationality | | | | | | | | | | | | |
| Qatari | 1.06 (0.82–1.37) | 0.636 | 1.11 (0.44–2.75) | 0.835 | 1.06 (0.96–1.18) | 0.249 | 1.11 (0.73–1.66) | 0.618 | 0.62 (0.35–1.1) | 0.097 | 1.23 (0.02–43.54) | 0.919 |
| Other Arabs | 0.89 (0.69–1.14) | 0.350 | 1.07 (0.44–2.59) | 0.878 | 0.82 (0.74–0.92) | **0.000** | 1.1 (0.75–1.6) | 0.620 | 0.41 (0.23–0.75) | **0.003** | 0.01 (0–2.81) | 0.106 |
| Other Nationalities | Ref | | Ref | | Ref | | Ref | | Ref | | Ref | |
| Education | | | | | | | | | | | | |
| Elementary and below | 0.85 (0.41–1.77) | 0.669 | 0.71 (0.16–3.11) | 0.661 | 0.85 (0.64–1.13) | 0.261 | 0.65 (0.32–1.27) | 0.213 | 4.41 (1.25–15.58) | **0.012** | 0.48 (0–63.5) | 0.773 |
| Secondary/Highschool | 1.06 (0.7–1.61) | 0.780 | 1.05 (0.47–2.29) | 0.912 | 0.92 (0.78–1.09) | 0.341 | 0.94 (0.64–1.35) | 0.742 | 2.26 (0.76–6.71) | 0.132 | 0.08 (0–3.2) | 0.182 |
| University or above | Ref | | Ref | | Ref | | Ref | | Ref | | Ref | |
| Diabetes Status | | | | | | | | | | | | |
| No DM | Ref | | Ref | | Ref | | Ref | | Ref | | Ref | |
| GDM | 0.84 (0.66–1.06) | 0.145 | 0.53 (0.21–1.29) | 0.163 | 1.19 (1.08–1.31) | **0.000** | 0.78 (0.53–1.12) | 0.183 | 0.96 (0.56–1.64) | 0.868 | 9.61 (0.37–125.31) | 0.172 |
| PGDM | 1.56 (0.92–2.64) | 0.098 | 2.4 (0.63–8.2) | 0.199 | 2.65 (2.24–3.14) | **0.000** | 2.48 (1.28–4.23) | **0.009** | 0.61 (0.08–4.4) | 0.618 | N/A | |
| Chronic Hypertension | | | | | | | | | | | | |
| No | Ref | | Ref | | Ref | | Ref | | Ref | | Ref | |
| Yes | 2.8 (1.64–4.78) | **0.000** | N/A | | 2.02 (1.57–2.6) | **0.000** | 1.27 (0.35–3.61) | 0.702 | N/A | | N/A | |
| Pre-or early-pregnancy BMI | | | | | | | | | | | | |
| Normal | Ref | | Ref | | Ref | | Ref | | Ref | | Ref | |
| Underweight | 0.7 (0.22–2.2) | 0.533 | 0.84 (0.16–4.08) | 0.840 | 1.54 (1.05–2.25) | **0.031** | 1.63 (0.76–3.13) | 0.203 | 1.56 (0.2–12.39) | 0.672 | N/A | |
| Overweight | 0.71 (0.47–1.07) | 0.099 | 0.53 (0.2–1.35) | 0.182 | 1.11 (0.93–1.33) | 0.246 | 1.17 (0.78–1.71) | 0.452 | 0.88 (0.32–2.43) | 0.807 | 0.08 (0–8.02) | 0.281 |
| Obese | 0.83 (0.56–1.22) | 0.339 | 1 (0.4–2.46) | 1.000 | 1.3 (1.09–1.55) | **0.003** | 1.13 (0.72–1.72) | 0.586 | 1.52 (0.62–3.7) | 0.356 | 7.08 (0.11–162.88) | 0.352 |
| Baby gender | | | | | | | | | | | | |

(*Continued*)

**Table 3.** (*Continued*)

| | Apgar <7 at 1 min | | | | NICU/death in LR/OT | | | | In–hospital mortality | | | |
|---|---|---|---|---|---|---|---|---|---|---|---|---|
| | cRR (95% CI) | *p* value | aRR (95% CI) | *p* value | cRR (95% CI) | *p* value | aRR (95% CI) | *p* value | cRR (95% CI) | *p* value | aRR (95% CI) | *p* value |
| Male | 1.31 (1.07–1.62) | **0.009** | 0.96 (0.47–1.94) | 0.919 | 1.29 (1.18–1.41) | **0.000** | 1.3 (0.95–1.77) | 0.100 | 0.9 (0.56–1.46) | 0.672 | 0.2 (0.01–4.19) | 0.304 |
| Female | Ref | | Ref | | Ref | | Ref | | Ref | | Ref | |
| Chromosomal/Congenital abnormalities | | | | | | | | | | | | |
| No | Ref | | Ref | | Ref | | Ref | | Ref | | Ref | |
| Yes | 9.73 (7.37–12.86) | **0.000** | 7.68 (2.63–18.64) | **0.000** | 8.25 (7.71–8.84) | **0.000** | 8.58 (7.06–9.28) | **0.000** | 94.1 (58.72–150.81) | **0.000** | 447.22 (7.45–599.54) | **0.007** |
| **Table 3–B: LGA[b] vs. AGA** | | | | | | | | | | | | |
| LGA | 1.5 (1.13–1.98) | **0.004** | 0.54 (0.18–1.65) | 0.287 | 1.52 (1.36–1.7) | **0.000** | 1.15 (0.78–1.66) | 0.483 | 0.84 (0.35–1.99) | 0.688 | N/A | |
| AGA | Ref | | Ref | | Ref | | Ref | | Ref | | Ref | |
| Gestational Age* | | | | | | | | | | | | |
| Preterm | 9.79 (8.03–11.95) | **0.000** | 15.14 (7.85–26.59) | **0.000** | 7.05 (6.54–7.6) | **0.000** | 5.75 (4.52–7.04) | **0.000** | 19.94 (12.08–32.91) | **0.000** | 629.38 (0–0) | 0.982 |
| Term | Ref | | Ref | | Ref | | Ref | | Ref | | Ref | |
| Maternal Age | | | | | | | | | | | | |
| Young adults (20–34 years) | Ref | | Ref | | Ref | | Ref | | Ref | | Ref | |
| Adolescents (<20 years) | 1.44 (0.79–2.6) | 0.230 | N/A | | 1.01 (0.75–1.36) | 0.956 | 0.3 (0.04–1.94) | 0.225 | 0.78 (0.11–5.67) | 0.809 | N/A | |
| Advanced maternal age (≥35 years) | 1.17 (0.92–1.5) | 0.210 | 1.08 (0.41–2.78) | 0.878 | 1.23 (1.11–1.36) | **0.000** | 1.66 (1.2–2.25) | **0.003** | 1.82 (1.09–3.05) | **0.021** | 7.55 (0.43–93.11) | 0.167 |
| Parity | | | | | | | | | | | | |
| Nulliparous | Ref | | Ref | | Ref | | Ref | | Ref | | Ref | |
| Parity ≥1 | 0.46 (0.38–0.57) | **0.000** | 0.83 (0.35–1.88) | 0.651 | 0.64 (0.59–0.7) | **0.000** | 0.59 (0.41–0.83) | **0.002** | 0.88 (0.52–1.48) | 0.621 | 0.48 (0.02–10.86) | 0.647 |
| Nationality | | | | | | | | | | | | |
| Qatari | 1.06 (0.82–1.37) | 0.636 | 1.14 (0.37–3.35) | 0.826 | 1.06 (0.96–1.18) | 0.249 | 1.14 (0.75–1.65) | 0.534 | 0.62 (0.35–1.1) | 0.097 | N/A | |
| Other Arabs | 0.89 (0.69–1.14) | 0.350 | 1.7 (0.66–4.28) | 0.274 | 0.82 (0.74–0.92) | **0.000** | 1.04 (0.73–1.48) | 0.804 | 0.41 (0.23–0.75) | **0.003** | 0.19 (0.01–2.64) | 0.216 |
| Other Nationalities | Ref | | Ref | | Ref | | Ref | | Ref | | Ref | |
| Diabetes Status | | | | | | | | | | | | |
| No DM | Ref | | Ref | | Ref | | Ref | | Ref | | Ref | |
| GDM | 0.84 (0.66–1.06) | 0.145 | 0.52 (0.19–1.35) | 0.179 | 1.19 (1.08–1.31) | **0.000** | 0.92 (0.66–1.27) | 0.601 | 0.96 (0.56–1.64) | 0.868 | 1.3 (0.06–24.49) | 0.868 |
| PGDM | 1.56 (0.92–2.64) | 0.098 | 2.9 (0.84–8.92) | 0.091 | 2.65 (2.24–3.14) | **0.000** | 1.85 (1.03–3.1) | **0.041** | 0.61 (0.08–4.4) | 0.618 | N/A | |
| Pre-or early-pregnancy BMI | | | | | | | | | | | | |
| Normal | Ref | | Ref | | Ref | | Ref | | Ref | | Ref | |
| Underweight | 0.7 (0.22–2.2) | 0.533 | 2.37 (0.47–9.92) | 0.291 | 1.54 (1.05–2.25) | **0.031** | 1.46 (0.61–3.09) | 0.385 | 1.56 (0.2–12.39) | 0.672 | N/A | |
| Overweight | 0.71 (0.47–1.07) | 0.099 | 0.95 (0.37–2.37) | 0.912 | 1.11 (0.93–1.33) | 0.246 | 1.11 (0.75–1.6) | 0.606 | 0.88 (0.32–2.43) | 0.807 | 1.38 (0.06–28.91) | 0.842 |
| Obese | 0.83 (0.56–1.22) | 0.339 | 0.88 (0.31–2.44) | 0.808 | 1.3 (1.09–1.55) | **0.003** | 1.21 (0.81–1.78) | 0.355 | 1.52 (0.62–3.7) | 0.356 | 2.41 (0.11–45.09) | 0.579 |
| Baby Gender | | | | | | | | | | | | |
| Male | 1.31 (1.07–1.62) | **0.009** | 0.87 (0.4–1.85) | 0.726 | 1.29 (1.18–1.41) | **0.000** | 1.24 (0.92–1.65) | 0.164 | 0.9 (0.56–1.46) | 0.672 | 1.36 (0.12–14) | 0.802 |

(*Continued*)

**Table 3.** (Continued)

| | Apgar <7 at 1 min | | | | NICU/death in LR/OT | | | | In–hospital mortality | | | |
|---|---|---|---|---|---|---|---|---|---|---|---|---|
| | cRR (95% CI) | p value | aRR (95% CI) | p value | cRR (95% CI) | p value | aRR (95% CI) | p value | cRR (95% CI) | p value | aRR (95% CI) | p value |
| Female | Ref | | Ref | | Ref | | Ref | | Ref | | Ref | |
| Employment Status | | | | | | | | | | | | |
| Employed | Ref | | Ref | | Ref | | Ref | | Ref | | Ref | |
| Unemployed | N/A | | N/A | | 0.68 (0.26–1.76) | 0.413 | 1.38 (0.39–4.02) | 0.607 | N/A | | N/A | |

Abbreviations: cRR, crude risk ratio; aRR, adjusted risk ratio; CI, confidence interval; Ref, referent; NA, not applicable; DM, diabetes mellitus; GDM, gestational diabetes mellitus; PGDM, pre-gestational diabetes mellitus; AGA, appropriate for gestational age; SGA, small for gestational age; LGA, large for gestational age; Apgar, Appearance, Pulse, Grimace, Activity, and Respiration; NICU, neonatal intensive care unit; LR, labor room; OT, operation theatre.

[a] adjusted for the risk factors associated with SGA that were significant in the univariate analysis, with *p*-values <0.05 (Table 2): Gestational age, maternal age, parity, nationality, education, diabetes status, chronic hypertension, early- or pre-pregnancy BMI, baby gender, and any chromosomal or congenital abnormalities (Table 3A).

[b] adjusted for the risk factors associated with LGA that were significant in the univariate analysis, with *p*-values <0.05 (Table 2): Gestational age, maternal age, parity, nationality, diabetes status, early- or pre-pregnancy BMI, baby gender, and employment status (Table 3B).

The variable Apgar<7 at 5 mins is not shown in this table due to missing data but shown in S4 Table. The full results are provided in S4 Table.

*The analysis presented here was conducted using the gestational age variable categorized into two groups. The full analysis was reconducted with gestational age categorized into five groups and is presented in S5 Table.

**Bold** values denote statistical significance at the p<0.05 level.

adjustment in the multivariate analysis, only extreme to very preterm (aRR, 54.67; 95% CI, 30.85–65.65) remained significantly associated with low Apgar score <7 at 1 minute (S5 Table). For in hospital mortality, extreme to very preterm (cRR, 102.37; 95% CI, 49.57–211.4), moderate preterm (cRR, 39.94; 95% CI, 15.4–103.59), and late preterm (cRR, 11.69; 95% CI, 5.14–26.59) were significantly associated with in hospital mortality but these were not applicable when adjusting for all confounders due to missing data. Finally, for NICU admission or death at LR or OT, extreme to very preterm (cRR, 15.14; 95% CI, 14.01–16.37), moderate preterm (cRR, 14.45; 95% CI, 13.27–15.74), late preterm (cRR, 5.62; 95% CI, 5.03–6.28), and early term (cRR, 1.41; 95% CI, 1.25–1.58) were significantly associated with higher risk of NICU admission or death at LR or OT. When adjusting for the confounding factors, moderate preterm (aRR, 14.39; 95% CI, 10.22–15.11), late preterm (aRR, 7.49; 95% CI, 5.55–9.45) and early term (aRR, 2.16; 95% CI, 1.48–3.06) remained significantly associated with high risk of NICU admission/death in LR/OT. Data for extremely to very preterm was not applicable due to missing data when adjusting for all confounders.

SGA was significant in all univariate analyses but became non-significant (or not applicable) when adjusting for the confounders except for NICU admission/death in LR/OT where it remained significant (S5 Table). The same analysis was performed for LGA where similar results were found except that in the multivariate analysis all preterm groups became non-significant for Apgar score <7 at 1 minute and that LGA became non-significant in all multivariate analysis (S5 Table).

Kaplan-Meier analyses was also performed to investigate the risk stratified algorithms. The analysis showed significant differences among the three groups (SGA, LGA and AGA) in incidence of low Apgar score at 1 minute, NICU/death in LR/OT and in-hospital mortality during the course of 24–40 weeks of gestation. Low Apgar score at 1 minute was observed in 0.8% of the AGA, 0.7% of the LGA, and 7.8% for SGA ($\chi^2$ (2, 14,601) = 142.92; P < 0.001) (Fig 1A). Admission to the NICU/death in LR/OT was observed in 9.8% of the AGA, 14.9% of the LGA, and 28.9% of the SGA ($\chi^2$ (2, 14,640) = 351.01; P < 0.001) (Fig 1B). In-hospital mortality was

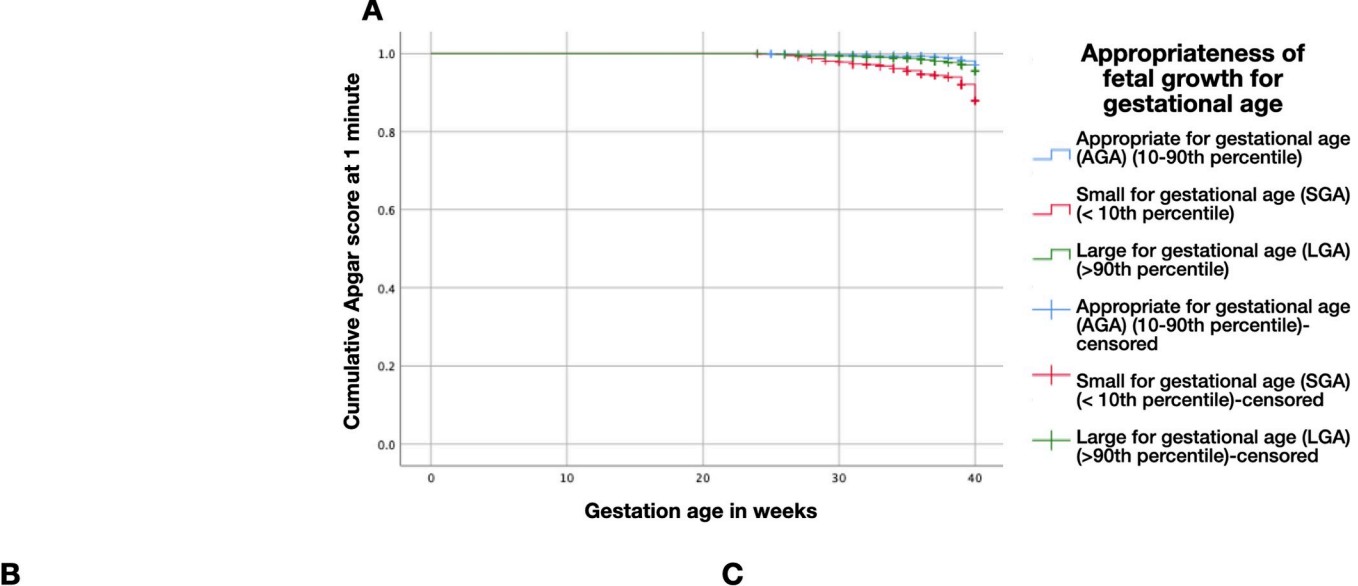

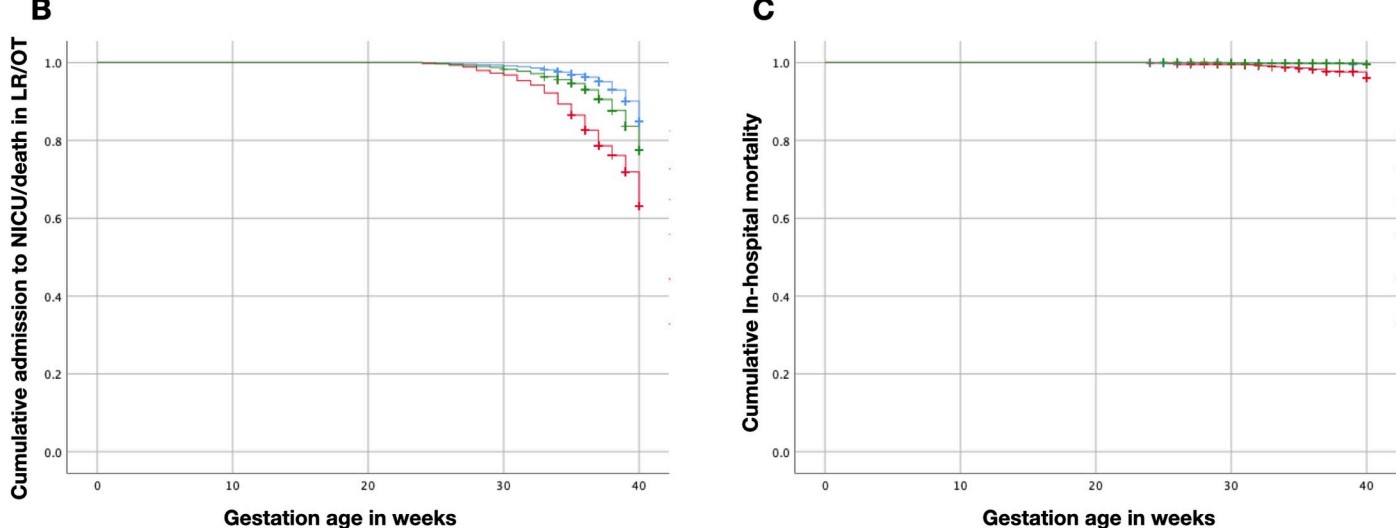

**Fig 1. Kaplan-Meier curves assessing differences in medians, among the three groups (AGA, SGA and LGA) for the outcomes during the course of 24–40 weeks of gestation.** (**A**) Apgar score, (**B**) NICU/death in LR/OT, and (**C**) in-hospital mortality.

observed in 0.3% of the AGA and LGA and 2.5% for SGA ($\chi^2$ (2, 14,641) = 98.08; P < 0.001) (Fig 1C).

## 4. Discussion

This large population-based study is the first of its kind to assess the incidence, maternal risk factors and neonatal outcomes associated with SGA and LGA in Qatar. A total of 14,641 singleton births registered in the PEARL database from April 2017 to March 2018 were examined. Our population-based study showed an SGA incidence of 60 per 1000 total singleton births (6.0%), which was relatively lower than previously reported in other countries [52–58]. Recently, a prospective cohort data between 1983 and 2006 conducted among 75,296 infants from 12 European countries revealed an SGA prevalence ranging from 4.6% in Finland up to 15.3% in Portugal [59]. On the other hand, the LGA incidence was estimated to be 156 per

1000 total singleton births (15.6%), which was comparable to previous reports in Vietnam [56] and Thailand [60].

The main reason behind such disparities could be mainly due to differences in the characteristics of the study populations, especially ethnic origins, race, and dietary habits. Furthermore advanced antenatal care at our institution [61], high quality counselling and support could have minimized SGA incidence in this population. In addition, it is noteworthy to mention that the present study comprised a total of 64.79% overweight/obese women (32.45% overweight and 32.34% obese), which is remarkably higher than the overweight/obesity average incidence rates around the globe [WHO Global Health Observatory: share of adults that are overweight or obese in 2016: Americas (62.5%), Europe (58.7%), Eastern Mediterranean (49%), Western Pacific (31.7%), Africa (31.1%), South-East Asia (21.9%)] [62]. This could have contributed to the high LGA and comparably low SGA incidence rates in Qatar compared to global estimates. According to a meta-analysis by Gaudet et al. [63], maternal obesity is significantly associated with the development of fetal overgrowth, with an 142% increase in the odds of delivering LGA among obese women compared with their normal weight counterparts. Furthermore, it has been reported that the percentage of LGA infants was significantly higher among overweight women even in the absence of GDM [64]. These findings indicate that being overweight and obesity are both determinants of fetal growth regardless of the presence of other risk factors. Moreover, diabetes, which is also believed to be strongly associated with fetal growth [65], was found to be very high in our sample population (Total DM: 31.68%; GDM: 29.09% and PGDM: 2.60%), which is relatively high compared to the rest of the world. For instance, according to 2019 estimates from the International Diabetes Federation, the average diabetes prevalence was estimated to be 15.33% in the Pacific island small states, 11.37% in Middle East & North Africa, 11.24% in South Asia, 10.46% in North America, and lower than 10% in Latin America and Caribbean, East Asia and Pacific, countries of the European Union, and Sub-Saharan Africa [66].

Furthermore, it is worth noting that most studies have advocated the use of local or customized charts to estimate SGA and LGA in particular populations [35, 36]; however, these local charts are only relevant to the population and time from which they were derived and making comparison between populations and studies impossible, and thus limits generalisability to other populations. However, in the present study, SGA and LGA estimates were calculated based on the multinational recently released INTERGROWTH-21st standard, which offers a reliable multinational tool for estimating fetal weight percentiles [34].

In our population-based study, preterm birth was significantly associated with SGA and LGA, with male infants significantly less likely to be SGA but high likely to be LGA (Table 2). In addition, parity≥1 was significantly associated with a low risk of SGA but a high risk of LGA (Table 2). Moreover, infants born to overweight mothers were significantly less likely to be born SGA. Further, GDM and PGDM were significantly associated with LGA births (Table 2). Several sociodemographic factors were found to be significantly associated with inappropriate birth weight for gestational age, including nationality which was significantly associated with LGA births in the adjusted model. Further, unemployment was found to be independently associated with LGA births. These are all well-established risk factors for SGA and LGA among different racial and ethnic groups [67–70]. Nevertheless, in contrast with other studies, consanguinity, smoking, and preterm history had no effect on SGA or LGA in the univariate and the multivariate analyses.

Over the past four decades, there has been a tremendous improvement in perinatal care, which has significantly improved the survival of infants born with low birth weight [71]. SGA infants were found to be at higher mortality risk than non-SGA infants or infants born within the normal weight span [72, 73]. Despite the main focus of research being on low birth weight,

a growing evidence suggests that there are existing U-shaped associations, with high birth weight linked to increased mortality risks [31–33]. To date, most studies investigating this area of research have primarily focused on investigating the link between birth weight and gestational age as separate components. A Swedish medical birth registry-based study has shown high mortality in individuals born early term [74]. In our study we found that NICU/death was 9.8% for AGA, 14.9% for LGA, and 28.9% for SGA (Table 1 and Fig 1). In-hospital mortality and admission to NICU/or death in LR/OT were significantly more likely to occur among SGA infants in comparison to AGA infants (Table 3). Furthermore, both SGA and LGA were significantly related to caesarean deliveries (Table 3). It is important to mention that while caesarean sections can be protective, they can lead to significant morbidities among both the mothers and their babies, and thus, the ideal delivery mode for SGA and LGA singletons remains controversial, particularly in preterm delivery cases [75].

Our study has several strengths. First of all, previous studies on this topic have investigated the risk factors and outcomes associated with birth weight and gestational age separately, but only few studies have looked at the risk factors and outcomes of birth weight in the context of gestational age, particularly LGA. The LGA group is a relatively new area of investigation, since most studies to date have focused on low birth weight, and only few reports have shown associations between high birth weight and increased mortality [31–33]. Moreover, in our study we also looked at the various categorisation of GA, there are very few studies that looked at extreme to very preterm, moderate preterm, late preterm, and early term in comparison to full term as we did in the present study. Furthermore, we were able to adjust for several demographic and medical confounding factors, known to affect fetal growth. It is generally recognized that inappropriate birth weight for gestational age is confounded by many factors, and published studies are very limited, particularly for LGA. So far, only few studies have determined PAFs for SGA and LGA, particularly, in the presence of confounding factors. Since unadjusted PAFs may be falsely high or low if confounding is present, the ability to adjust for relevant confounders and calculate multivariable-adjusted PAFs is another strength of the current analysis. We provided information on the population burden of SGA and LGA due to the underlying risk factors by determining the adjusted PAFs. The adjusted PAFs in the current paper can help our understanding of the extent to which SGA and LGA can be reduced if the assessed risk factors were eliminated. It gives a percentage of reducing the risk and improving the protective factors. Finally, this study used data from the PEARL-Peristat Study. The PEARL-Peristat Study is an ongoing cohort study based on the predesigned hospital data pertaining to mothers and their newborns. In its initial phase, the PEARL study was conducted from 2011 to 2013, while this phase covered the 2017–2019 period [76]. This registry reports data on maternal, neonatal and perinatal mortality, morbidities, and their correlates, including data on live births and neonatal mortality from all public and private maternity facilities in Qatar [76, 77]. This database is large enough with a sample size that is generally representative of births in Qatar. In addition, HMC is the main national hospital, the main provider of secondary and tertiary healthcare in Qatar, consisting of multiple regional hospitals that are widely distributed in different geographical areas of Qatar, and account for the majority of births in the country. Furthermore, selection bias was minimized via examining all available live births for the study period.

Despite being the largest study of its kind in the State of Qatar, this study has some limitations. Although we carefully adjusted for several potential confounders, we were unlikely to fully rule out the possibility of residual confounding. Thus, it is noteworthy to mention that the observed associations might be attributable to unmeasured confounders such as parents' history of SGA or LGA births. In addition, there were missing data on some variables, which were excluded from the analysis. However, the missing data in each of these variables were

comparable across the subgroups, therefore these missing data are unlikely to have affected our reported estimates. However, the sample size for some factors were very small (e.g., mortality amongst SGA = 22/882 (2.5%), which could have caused an overestimated RR, particularly after adjusting for confounding factors. In addition, empirical evidence indicates that the validity of regression models is only slightly affected after selective dropout. Thus, the relation between risk factors and outcome is unlikely to be considerably changed by selective dropout [78]. Our results therefore support the evidence on the association between different risk factors and fetal growth.

## 5. Conclusion

This is the first population-based study to assess the incidence, risk factors and feto-maternal outcomes associated with inappropriate fetal growth in Qatar. In summary, the present study identified several risk factors that are associated with SGA and LGA births, including maternal medical and social conditions. In addition, prematurity was found to be significantly associated with SGA and LGA births, with male infants being less likely to be born SGA but high likely to be born LGA, in comparison to female infants. Moreover, SGA increased the risk of neonatal mortality and admission to NICU, as well as death in labor room and operation theatre. It is noteworthy to mention that many of the identified risks are potentially modifiable (e.g., maternal medical conditions, or lifestyle habits), suggesting avenues for possible prevention of SGA and LGA in future pregnancies. Modifiable risk factors should be identified as early as possible and managed accordingly. Thus, perinatal monitoring and antenatal care are essential to reduce the burden of inappropriate fetal growth and increase the chance of survival.

## Supporting information

**S1 Table. Crosstabs of the associated risk factors and outcomes.**
(XLSX)

**S2 Table. Univariate and multivariate regression of the risk factors associated with SGA and LGA births, with gestational age categorized into two groups (preterm vs. term).**
(XLSX)

**S3 Table. Univariate and multivariate regression of the risk factors associated with SGA and LGA births, with gestational age categorized into five groups (extreme to very preterm, moderate preterm, late preterm, early term, and full term).**
(XLSX)

**S4 Table. Univariate and multivariate regression of the pregnancy and neonatal outcomes associated with SGA and LGA births, with gestational age categorized into two groups (preterm vs. term).**
(XLSX)

**S5 Table. Univariate and multivariate regression of the pregnancy and neonatal outcomes associated with SGA and LGA births, with gestational age categorized into five groups (extreme to very preterm, moderate preterm, late preterm, early term, and full term).**
(XLSX)

## Acknowledgments

The authors want to thank their respective institutions for their continued support.

## Author Contributions

**Conceptualization:** Salma Younes, Muthanna Samara, Noor Salama, Rana Al-jurf, Tawa Olukade, Sara Hammuda, Ghassan Abdoh, Palli Valapila Abdulrouf, Thomas Farrell, Hilal Al Rifai, Nader Al-Dewik.

**Data curation:** Salma Younes, Noor Salama, Tawa Olukade, Sara Hammuda, Ghassan Abdoh, Palli Valapila Abdulrouf, Thomas Farrell, Hilal Al Rifai.

**Formal analysis:** Salma Younes, Noor Salama, Ghassan Abdoh, Palli Valapila Abdulrouf, Thomas Farrell, Hilal Al Rifai, Nader Al-Dewik.

**Funding acquisition:** Salma Younes, Muthanna Samara.

**Investigation:** Sawsan Al-Obaidly, Ghassan Abdoh, Hilal Al Rifai.

**Methodology:** Gheyath Nasrallah, Ghassan Abdoh, Hilal Al Rifai.

**Validation:** Muthanna Samara.

**Writing – original draft:** Salma Younes, Nader Al-Dewik.

**Writing – review & editing:** Salma Younes, Muthanna Samara, Noor Salama, Rana Al-jurf, Gheyath Nasrallah, Sawsan Al-Obaidly, Husam Salama, Tawa Olukade, Sara Hammuda, Ghassan Abdoh, Palli Valapila Abdulrouf, Thomas Farrell, Mai AlQubaisi, Hilal Al Rifai, Nader Al-Dewik.

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
