## [Decision Letter · Decision Letter 0]

17 May 2021

PONE-D-21-07926

Incidence, Risk Factors, and Feto-Maternal Outcomes of Inappropriate Birth Weight for Gestational Age: A Population-Based Study

PLOS ONE

Dear Dr. Al-Dewik,

Thank you for submitting your manuscript to PLOS ONE. After careful consideration, we feel that it has merit but does not fully meet PLOS ONE’s publication criteria as it currently stands. Therefore, we invite you to submit a revised version of the manuscript that addresses the points raised during the review process.

The reviewers' concerns on methodological aspects need to be addressed.

We look forward to receiving your revised manuscript.

Kind regards,

Dayana Farias, Ph.D

Academic Editor

PLOS ONE

Journal Requirements:

2. For more information on PLOS ONE's expectations for statistical reporting, please see https://journals.plos.org/plosone/s/submission-guidelines.#loc-statistical-reporting. Please update your Methods and Results sections accordingly.

3.Please note that in order to use the direct billing option the corresponding author must be affiliated with the chosen institute. Please either amend your manuscript to change the affiliation or corresponding author, or email us at plosone@plos.org with a request to remove this option.

4.Your ethics statement should only appear in the Methods section of your manuscript. If your ethics statement is written in any section besides the Methods, please delete it from any other section.

Reviewers' comments:

Reviewer's Responses to Questions

**Comments to the Author**

1. Is the manuscript technically sound, and do the data support the conclusions?

Reviewer #1: Yes

Reviewer #2: No

2. Has the statistical analysis been performed appropriately and rigorously? 

Reviewer #1: Yes

Reviewer #2: I Don't Know

3. Have the authors made all data underlying the findings in their manuscript fully available?

Reviewer #1: No

Reviewer #2: No

4. Is the manuscript presented in an intelligible fashion and written in standard English?

Reviewer #1: Yes

Reviewer #2: Yes

5. Review Comments to the Author

Reviewer #1: General comments

- The study would be of greater contribution if provides the performance of risk assessment models based on the significant risk factor. Risk-stratified algorithms are of great interest in clinical practice. Additionally, consider providing population prevalence and population attributable fractions for the associated factors.

- Detailed description of the method for estimating gestational age is crucial. It is strongly recommended to report (Suppl Info) the proportion of women whose gestational age was estimated by each method – for instance, by 1) LMP only; 2) early US only; 3) LMP and early US; 4) LMP and late US; or 5) late US only.

- Provide absolutes risks for the outcomes.

Title

- Make clear what is the setting of the study; which population is this study was related to?

Abstract

- (Background) Preferably, use inadequate than aberrant.

- What do you mean by “long-term complication during pregnancy”?

- Describe the reference chart applied for classifying adequacy of birth weight.

- The sentence “Preterm birth significantly predicted SGA and LGA” did not seem accurate. Prediction implies diagnostic performance assessment, which was not the case in this analysis. Please, revise. Same for GDM and PGDM significantly predicted LGA.

- Avoid using abbreviations in the abstract, especially when they were not previously detailed (e.g., PGDM and LR/OT).

- How could SGA and LGA lead to c-section if they are defined by a postnatal condition (birth weight)?

- Conclusion: The fact the this was the first study in Qatar does not necessarily novel contributions to the research field. In addition, it is not a reasonable conclusion foe the abstract.

Methods

- Consider using SGA- and LGA-related outcomes. Precious studies have shown that SGA and LGA may impact differently on perinatal outcomes (e.g. Neonatal Morbidity of Small- and Large-for-Gestational-Age Neonates Born at Term in Uncomplicated Pregnancies. Obstet Gynecol. 2017;130(3):511-9).

- Describe in more detail the tool used to estimate birth weight centiles. Several studies conducted in different populations concluded that chart-specific thresholds for a specific population should be considered in clinical practice, once different charts have different performance on identifying SGA and LGA babies. Why choosing this specific tool (Ref 34)?

- Definition of some variables are not clear enough. Please, make clear the definition for smoking (Have cessation during pregnancy been considered?), preterm history (any PTB or only spontaneous?), GDM (IADPSG criteria? ADA criteria? Local criteria?), maternal BMI (Self-reported weight? Have you also considered early pregnancy weight?).

- The risks for adverse outcomes according to gestational age may be different for the early, full and late terms. When calculating the risk ratios, consider using the three categories instead of only newborns delivered between 39 and 40 weeks.

- Birth weight categorized as low and normal seem useless.

Results

- Why using Odds ration instead of relative risk. Preferably, use aRR.

- Provide absolutes risks for the outcomes.

Reviewer #2: Thanks for the opportunity to review this manuscript where the aim was to assess the incidence, risk factors, and feto-maternal outcomes associated with small-forgestational age (SGA) and large-for-gestational age (LGA) infants.

Overall: It is well known both that SGA, LGA and gestational week will be associated to neonatal outcome. Even if this is the largest study made in Qatar, there are other studies with the same aim. Please clarify the novelty of this study and the reason it adds new knowledge.

Abstract:

1. For example, in this sentence and throughout the manuscript please use the word associations instead of predicted/predictors. Because what you have tested is the potential association, you havn’t performed any predictive models. A variable will not necessarily be a good predictor just because there is a significant association “GDM (aOR 1.45, 95% CI:1.13–1.86) and PGDM (aOR 3.51, 95% CI:2.08–5.92) significantly predicted LGA.”

2. “Both SGA (aOR, 1.56; 95% CI:1.06–2.31) and LGA (aOR, 1.34, 95% CI:1.04–1.73) significantly lead to caesarean deliveries”, According to the previous reasoning, SGA and LGA was associated to cesarean deliveries, it is hard from observational studies to be able to demonstrate causal relationships.

Background:

3. Gestational age is a strong indicator (??) of birthweight and fetal growth, both of which are influenced by a combination of environmental and genetic influences

Material and methods:

4. The study is based on 14000 deliveries from one large hospital, please give us some more information about the population to make it easier to evaluate the external validity of the study to other populations.

5. What was gestational age based on? Ultrasound in gw 10-12? Ultrasound in gw 18-20? Period data?

6. Throughout the world we use different scales fetal growth, hence SGA is not always the same in different countries. What is you definition of AGA/SGA/LGA based on? Which algorithm/model?

7. What was the definition of GDM? Was OGTT done? Which glucose values were defined as GDM?

8. You write that you adjust the models for all significant predictors. There are potential confounders and potential mediators, however you have not done predicting models and therefor it is not possible to call the variables predictors. Some of the variables you adjust for, as gestational age and fetal sex for example will not be potential confounder, they will be potential mediators, meaning that a part of the effect on SGA/LGA will pass through that mediator. The mediator could potentially explain a part of the association.

9. Please make the exposures in the regression clear, it is a bit hard to follow the reasoning.

Results:

10. Please change in the results so that confounders/mediators and so on is coherent.

Discussion

11. You write “we took this a step further and combined gestational age with weight”. Please explain in what way you did this.

Conclusion

12. In the conclusion you write: “The findings of this study should be applied in antenatal care; in particular nutritional surveillance, support, and monitoring should be controlled to reduce the burden of inappropriate fetal growth.” The results of your study are associations to SGA/LGA neonatals. That there are associations does not mean that we know the cause behind the associations or that we know what to do to decrease risks. Please keep the conclusion to what you have studied.

6. PLOS authors have the option to publish the peer review history of their article (what does this mean?). If published, this will include your full peer review and any attached files.

Reviewer #1: No

Reviewer #2: No

---

## [Author Response · Author response to Decision Letter 0]

3 Jul 2021

Point by point response

PONE-D-21-07926

Incidence, Risk Factors, and Feto-Maternal Outcomes of Inappropriate Birth Weight for Gestational Age: A Population-Based Study

PLOS ONE

We would like to take this opportunity to thank the editor and the reviewers for their comments and for taking the time to evaluate our manuscript. We have revised the article accordingly and point-by-point responses to the comments are reported below. All the changes made to the manuscript are highlighted in the Main Document using “Track Changes” tool.

Please note that other reviewers have requested other changes, and thus these will also appear in the manuscript.

Please follow the clean version of the manuscript with track changes turned “Off” for the indicated line numbers.

Reviewer #1: General comments

- The study would be of greater contribution if provides the performance of risk assessment models based on the significant risk factor. Risk-stratified algorithms are of great interest in clinical practice. Additionally, consider providing population prevalence and population attributable fractions for the associated factors.

Response: We have done all of these extra analyses, including risk stratified algorithm using Kaplan Meier analysis (please see statistical analysis subsection Page 11, lines 176-180, results section Page 29, lines 329-340). Population prevalence and population attributable fractions were also calculated and added to Table 2, Table S2, and Table S3 (please see statistical analysis subsection Page 11, lines 167-175, and results section Pages 16-19, and page 20, lines 254-261).

- Detailed description of the method for estimating gestational age is crucial. It is strongly recommended to report (Suppl Info) the proportion of women whose gestational age was estimated by each method – for instance, by 1) LMP only; 2) early US only; 3) LMP and early US; 4) LMP and late US; or 5) late US only.

Response: We agree about the need for a detailed description of the method for estimating gestational age. Unfortunately, the Pearl-Peristat Study is an observational study primarily designed for broader maternal and neonatal outcomes in the perinatal period using routinely collected hospital data. Thus, the data was not exhaustive and we do not have access to such granular data which could have added value to our findings. We have added this issue also as a study limitation.

- Provide absolutes risks for the outcomes.

Response: we thank the reviewer for this suggestion, we calculated the absolute risks for SGA and LGA (presented in Table S2 and Table S3), and for Apgar score, NICU/death in LR/OT, and in-hospital mortality (Table S4-A and Table S4-B).

Title

- Make clear what is the setting of the study; which population is this study was related to?

Response: We changed the title to clarify the study setting and the study population.

Abstract

- (Background) Preferably, use inadequate than aberrant.

Response: We changed the word aberrant to abnormal. As the study is on both small-for-gestational age (SGA) and large-for-gestational age born infants, and thus we thought that the word abnormal might reflect the main variables (Page 3, line 2).

- What do you mean by “long-term complication during pregnancy”?

Response: We rephrased the sentence and removed long term complications during pregnancy (Page 3, lines 2,3). 

- Describe the reference chart applied for classifying adequacy of birth weight.

Response: We added an explanation about the tool and methodology used to estimate the centiles and the adequacy of birth weight (Pages 7,8 lines 90-l03). 

- The sentence “Preterm birth significantly predicted SGA and LGA” did not seem accurate. Prediction implies diagnostic performance assessment, which was not the case in this analysis. Please, revise. Same for GDM and PGDM significantly predicted LGA.

Response: We changed the words “predicted” / “predict” to “associated with” across the manuscript.

- Avoid using abbreviations in the abstract, especially when they were not previously detailed (e.g., PGDM and LR/OT).

Response: We replaced the indicated abbreviations with the full terms (Page 3, line 15-19).

- How could SGA and LGA lead to c-section if they are defined by a postnatal condition (birth weight)?

Response: We have removed the regression analysis of delivery mode (vaginal vs. cesarean), and removed it from the table of outcomes (Table 3). We have only left the descriptive statistics in Table 1. 

- Conclusion: The fact the this was the first study in Qatar does not necessarily novel contributions to the research field. In addition, it is not a reasonable conclusion foe the abstract.

Response: The indicated sentence has been removed from the abstract (Page 3).

Methods

- Consider using SGA- and LGA-related outcomes. Precious studies have shown that SGA and LGA may impact differently on perinatal outcomes (e.g. Neonatal Morbidity of Small- and Large-for-Gestational-Age Neonates Born at Term in Uncomplicated Pregnancies. Obstet Gynecol. 2017;130(3):511-9).

Response: We considered Apgar score, in-hospital mortality, admission to neonatal intensive care unit or death in labor room or operation theatre as outcomes, unfortunately we don’t have any other perinatal outcomes in the current paper. In future investigations, we will build up on the current analysis, and include more feto-maternal outcomes.

- Describe in more detail the tool used to estimate birth weight centiles. Several studies conducted in different populations concluded that chart-specific thresholds for a specific population should be considered in clinical practice, once different charts have different performance on identifying SGA and LGA babies. Why choosing this specific tool (Ref 34)?

Response: This tool provides a simple spreadsheet-based estimated fetal weight percentile calculator and corresponding R software package to encompass 6 fetal growth standards: the INTERGROWTH-21st, World Health Organization (WHO), National Institute of Child Health and Human Development (NICHD), Perinatology Research Branch (PRB/NICHD), and the Hadlock et al3 and Fetal Medicine Foundation (FMF) standards. These calculations take into account mother characteristics including ethnic/race group as well as height and weight. Our study included different ethnicities and nationalities. Thus, the tool we used takes into account the population (ethnic/race) that is investigated in the study. 

We have added this information on how we estimate birth weight centiles in the methodology (please see Pages 7,8, lines 90-103.

- Definition of some variables are not clear enough. Please, make clear the definition for smoking (Have cessation during pregnancy been considered?), preterm history (any PTB or only spontaneous?), GDM (IADPSG criteria? ADA criteria? Local criteria?), maternal BMI (Self-reported weight? Have you also considered early pregnancy weight?).

Response: We have put more details about some of the indicated variables including GDM Page 9, lines 124-134), maternal BMI (Page 9, lines 135-140). and preterm history (Page 10, lines 141,142). Unfortunately, we don’t have data for smoking cessation during pregnancy. 

- The risks for adverse outcomes according to gestational age may be different for the early, full and late terms. When calculating the risk ratios, consider using the three categories instead of only newborns delivered between 39 and 40 weeks.

Response: We calculated the relative risk ratios for “preterm” using “term” as a reference group, which included all deliveries at at 37 weeks’ and above (Page 8, lines 105-107).

We have done extra analysis and further categorized gestational age into: extreme-to-very preterm, moderate preterm, late preterm, and early term, vs. full term (Page 8, lines 105-109). Please see Tables S3 and S5, for the full univariate and multivariate analyses repeated with the gestational age groups categorized as indicated. The findings were incorporated into the results section, page 21, lines 262-272, and page 28, lines 304-322.

The percentiles calculated using the FETAL-GPSXL calculator, are up to 40 weeks only (280 days) (Page 7, line 92), it was not applicable to calculate the percentiles for “late term” from 41 weeks, 0 days' to 41 weeks, 6 days' gestation, and “post-term” at 42 weeks'.

- Birth weight categorized as low and normal seem useless.

Response: we agree with the reviewer, we deleted it from the neonatal factors.

Results

- Why using Odds ration instead of relative risk. Preferably, use aRR.

Response: We converted all ORs to RRs and presented them in Table 2, Table 3, and the supplementary materials (Table S2, Table S3, Table S4, and Table S5). We rephrased the abstract and results accordingly.

- Provide absolutes risks for the outcomes.

Response: we thank the reviewer for this suggestion, we calculated the absolute risks for SGA and LGA (presented in Table S2 and Table S3), and for Apgar score, NICU/death in LR/OT, and in-hospital mortality (Table S4-A and Table S4-B).

Reviewer #2: Thanks for the opportunity to review this manuscript where the aim was to assess the incidence, risk factors, and feto-maternal outcomes associated with small-forgestational age (SGA) and large-for-gestational age (LGA) infants.

Response: we would like to take this opportunity to thank the editor and the reviewers for their comments and for taking the time to evaluate our manuscript. We have revised the article accordingly and point-by-point responses to the comments are reported below. All the changes made to the manuscript are highlighted in the Main Document using “Track Changes” tool.

Please note that other reviewers have requested other changes and thus these will also appear in the manuscript. 

Please follow the clean version of the manuscript with track changes turned “off” for the indicated line numbers.

Overall: It is well known both that SGA, LGA and gestational week will be associated to neonatal outcome. Even if this is the largest study made in Qatar, there are other studies with the same aim. Please clarify the novelty of this study and the reason it adds new knowledge.

Response: We have indicated in detail the novelty of our study in the introduction section (please see Page 6, lines 57-72) and discussion (please see Pages 32,33, lines 386-416). 

Abstract:

1. For example, in this sentence and throughout the manuscript please use the word associations instead of predicted/predictors. Because what you have tested is the potential association, you havn’t performed any predictive models. A variable will not necessarily be a good predictor just because there is a significant association “GDM (aOR 1.45, 95% CI:1.13–1.86) and PGDM (aOR 3.51, 95% CI:2.08–5.92) significantly predicted LGA.”

Response: We changed the words “predicted” / “predict” to “associated with” across all the manuscript.

2. “Both SGA (aOR, 1.56; 95% CI:1.06–2.31) and LGA (aOR, 1.34, 95% CI:1.04–1.73) significantly lead to caesarean deliveries”, According to the previous reasoning, SGA and LGA was associated to cesarean deliveries, it is hard from observational studies to be able to demonstrate causal relationships.

Response: We agree with the reviewer, we rephrased this section (Page 3, lines 11-20). The abstract was rephrased due to the changes made in the statistical analysis according to the recommendations indicated by the reviewer.

Background:

3. Gestational age is a strong indicator (??) of birthweight and fetal growth, both of which are influenced by a combination of environmental and genetic influences.

Response: We rephrased this section following the reviewer’s comment, and following the description in the Canadian Perinatal Health Report, 2000, 2003 (1, 2) (Page 5, lines 44-46).

Material and methods:

4. The study is based on 14000 deliveries from one large hospital, please give us some more information about the population to make it easier to evaluate the external validity of the study to other populations.

Response: We have added more information about the population and the hospitals involved in the study (Page 7, lines 80-88). 

5. What was gestational age based on? Ultrasound in gw 10-12? Ultrasound in gw 18-20? Period data?

Response: We have added the information about gestational age in the methodology section (please see Page 8, lines 104,105). 

6. Throughout the world we use different scales fetal growth, hence SGA is not always the same in different countries. What is you definition of AGA/SGA/LGA based on? Which algorithm/model?

Response: We used the FETALGPSXL tool. This tool provides a simple spreadsheet-based estimated fetal weight percentile calculator and corresponding R software package to encompass 6 fetal growth standards: the INTERGROWTH-21st, World Health Organization (WHO), National Institute of Child Health and Human Development (NICHD), Perinatology Research Branch (PRB/NICHD), and the Hadlock et al3 and Fetal Medicine Foundation (FMF) standards. These calculations take into account mother characteristics including ethnic/race group as well as height and weight. Our study included different ethnicities and nationalities. Thus, the tool we used takes into account the population (ethnic/race) that is investigated in the study. 

We have added this information on how we estimate birth weight centiles in the methodology (please see Pages 7,8, lines, 90-103).

7. What was the definition of GDM? Was OGTT done? Which glucose values were defined as GDM?

Response: We have added this information in the methodology section (please see Page 9, lines 124-134) 

8. You write that you adjust the models for all significant predictors. There are potential confounders and potential mediators, however you have not done predicting models and therefor it is not possible to call the variables predictors. Some of the variables you adjust for, as gestational age and fetal sex for example will not be potential confounder, they will be potential mediators, meaning that a part of the effect on SGA/LGA will pass through that mediator. The mediator could potentially explain a part of the association.

Response: We thank the reviewer for his/her pertinent comment and completely agree with his/her point. We changed the words “predicted” / “predict” to “associated with” across the manuscript.

We have also rephrased the results section to indicate that GA and gender are mediators while the other variables are confounders associated to SGA/LGA (please see results section). 

9. Please make the exposures in the regression clear, it is a bit hard to follow the reasoning.

Response: We have explained the regression analysis in the statistical analysis subsection in detail (please see Pages 10,11, lines 151-166). We have also rephrased the results section to further clarify the exposures (please see the results section). We added information to indicate in detail the adjusted factors (please see the footnotes of Table 2, Table 3, and also in the supplementary materials: Table S2, Table S3, Table S4, and Table S5).

Results:

10. Please change in the results so that confounders/mediators and so on is coherent.

Response: We have made the explanation in the results section clearer and coherent with regards to confounders and mediators (Page 15, 20, 21, and 28). We have also indicated this in the statistical analysis subsection (Page 10). 

Discussion

11. You write “we took this a step further and combined gestational age with weight”. Please explain in what way you did this.

Response: We agree that this sentence is not clear, and thus we have removed it. We have now explained in detail the tool and methodology on how we calculated the fetal growth according to gestational age (please see (Pages 7,8 lines 90-l03). We have also highlighted this in the introduction section (page 6, lines 57-60).

Conclusion

12. In the conclusion you write: “The findings of this study should be applied in antenatal care; in particular nutritional surveillance, support, and monitoring should be controlled to reduce the burden of inappropriate fetal growth.” The results of your study are associations to SGA/LGA neonatals. That there are associations does not mean that we know the cause behind the associations or that we know what to do to decrease risks. Please keep the conclusion to what you have studied.

Response: The conclusion section was rephrased as indicated (Pages 33,34; lines 428-434).

 

References

1. Health Canada. Canadian Perinatal Health Report, 2000. Ottawa, ON: Minister of Public Works and Government Services Canada, 2000.

2. Health Canada. Canadian Perinatal Health Report, 2003. Ottawa, ON: Minister of Public Works and Government Services Canada, 2003.

---

## [Decision Letter · Decision Letter 1]

4 Aug 2021

PONE-D-21-07926R1

Incidence, Risk Factors, and Feto-Maternal Outcomes of Inappropriate Birth Weight for Gestational Age Among Singleton Live Births in Qatar: A Population-Based Study

PLOS ONE

Dear Dr. Al-Dewik,

Thank you for submitting your manuscript to PLOS ONE. After careful consideration, we feel that it has merit but does not fully meet PLOS ONE’s publication criteria as it currently stands. Therefore, we invite you to submit a revised version of the manuscript that addresses the points raised during the review process.

We look forward to receiving your revised manuscript.

Kind regards,

Dayana Farias, Ph.D

Academic Editor

PLOS ONE

Journal Requirements:

Additional Editor Comments (if provided):

The authors have addressed almost all comments, but some clarification is still needed.

Does this method consider the sex of the newborn? If so, please include it on page 7 lines 91-92.

Page 8 line 108 the word preterm is missing “extreme to very... preterm”

Regarding mother age classification, the term “normal age” is not quite accurate. So, I suggest the use of young adults, adolescents, and advanced maternal age.

Page 10, Lines 148 and 149 have the same meaning, please keep only one version of it.

148 “All categorical and binary variables were expressed as numbers and percentages”

149 “Variables were summarized using numbers and percentages”

The definition of mediator factor is a variable that is in the middle of the causal path of exposure and outcome. SGA birth is not a cause of sex, for example. So, please check the classification of preterm birth and sex as mediators in the models.

Please check table 3 adjusted value of SGA for In-hospital mortality, the RR change from 7.95 (4.7– 13.46) to 226.56 (3.47– 318.22). Is it right? Which variables are causing this change?

Reviewers' comments:

Reviewer's Responses to Questions

**Comments to the Author**

1. If the authors have adequately addressed your comments raised in a previous round of review and you feel that this manuscript is now acceptable for publication, you may indicate that here to bypass the “Comments to the Author” section, enter your conflict of interest statement in the “Confidential to Editor” section, and submit your "Accept" recommendation.

Reviewer #1: All comments have been addressed

Reviewer #2: (No Response)

2. Is the manuscript technically sound, and do the data support the conclusions?

Reviewer #1: Yes

Reviewer #2: Partly

3. Has the statistical analysis been performed appropriately and rigorously? 

Reviewer #1: Yes

Reviewer #2: Yes

4. Have the authors made all data underlying the findings in their manuscript fully available?

Reviewer #1: No

Reviewer #2: No

5. Is the manuscript presented in an intelligible fashion and written in standard English?

Reviewer #1: Yes

Reviewer #2: Yes

6. Review Comments to the Author

Reviewer #1: - It is not clear which standardized birth weight chart was used to classify neonates into SGA, AGA and LGA. Why using six methods (charts) based on fetal growth/weight standards for classifying appropriateness of birth weight?

- Surprisingly, the method used for classifying adequacy of birth weight resulted in 6% of SGA and 16% of LGA neonates. Considering that the expected rate would be 10% for SGA and LGA, why did you find such discrepancy? What are the implications for 1) the interpretation of your findings and 2) the generalizability of your study?

- What contribution to clinical practice does your study give?

Reviewer #2: Thanks for the opportunity to re-review this manuscript where the aim was to assess the incidence, risk factors, and feto-maternal outcomes associated with small-forgestational age (SGA) and large-for-gestational age (LGA) infants.

The comments have been adressed, however it is still confusing and un-clear with the exposure and outcome in the different analyses. You write:“Firstly, logistic regression analysis was performed for risk factors/confounders (demographic and medical factors) and mediators (prematurity and gender) of appropriateness of fetal growth for the GA groups (SGA/LGA vs. AGA).”

As I understand it you have used SGA/LGA as outcome in these analysis and different risk factors as exposure. Which variables are confounders/mediators will depend on which variable is exposure and which one is the outcome. If SGA/LGA is the outcome preterm birth will be a mediator. However, in the analyses of outcome:

“Secondly, logistic regression was performed to investigate the outcomes of SGA and LGA including Apgar score, NICU/death in LR/OT, and in-hospital mortality. Multiple logistic regression was performed (including all significant confounders and mediators from the univariate analysis) to investigate the association of SGA/LGA with Apgar score, NICU/death in LR/OT, and in-hospital mortality as outcomes.”

I guess that in these analyses SGA/LGA were the exposure and then preterm birth has another role, in this setting preterm birth will be a confounder (or in some cases a mediator if we believe that the SGA/LGA caused the premature birth) Gender will be a mediator in the first analyses and a confounder in the second analyses. Please, make it clear what the exposure and the outcome is in the different analyses.

I note that attributable fractions are added. For example you write “not delivered preterm 11.6%, indicating that almost 12% of SGA cases could have been prevented if mothers had not delivered preterm”, it would be good to add, in the discussion part, something about possible unmeasured confounding.

The numbers in table 1, “pregnancy mode” look strange. The percentages do not add up to 100%.

In the conculsion you write: "SGA and LGA births are related multi-factor interactions of demographic and medical confounders that can be mediated by prematurity and gender of the baby". This sentence is un-clear. What do you mean? What do you mean with interactions?

7. PLOS authors have the option to publish the peer review history of their article (what does this mean?). If published, this will include your full peer review and any attached files.

Reviewer #1: No

Reviewer #2: No

---

## [Author Response · Author response to Decision Letter 1]

9 Sep 2021

Point by point response

PONE-D-21-07926R1

Incidence, Risk Factors, and Feto-Maternal Outcomes of Inappropriate Birth Weight for Gestational Age Among Singleton Live Births in Qatar: A Population-Based Study

PLOS ONE

We would like to take this opportunity to thank the editor and the reviewers for their comments and for taking the time to evaluate our manuscript. We have revised the article accordingly and point-by-point responses to the comments are reported below. All the changes made to the manuscript are highlighted in the Main Document using “Track Changes” tool.

Please note that other reviewers have requested other changes, and thus these will also appear in the manuscript.

Please follow the clean version of the manuscript with track changes turned “Off” for the indicated line numbers.

Journal Requirements:

Response: All references have been checked, all are complete and correct.

Additional Editor Comments (if provided):

The authors have addressed almost all comments, but some clarification is still needed.

Does this method consider the sex of the newborn? If so, please include it on page 7 lines 91-92.

Response: Yes, the method considers newborn sex, the suggested change has been made (page 8, line 105).

Page 8 line 108 the word preterm is missing “extreme to very... preterm”

Response: The suggested change has been made, the word “preterm has been added (page 8, line 118).

Regarding mother age classification, the term “normal age” is not quite accurate. So, I suggest the use of young adults, adolescents, and advanced maternal age.

Response: The suggested changes have been made, the terms have been changed in the text (page 9, line 126; page 15, line 211, page 15, line 220, page 21, line 258, main tables (Tables 1-3), and supplementary materials (Tables S1-S5).

Page 10, Lines 148 and 149 have the same meaning, please keep only one version of it.

148 “All categorical and binary variables were expressed as numbers and percentages”

149 “Variables were summarized using numbers and percentages”

Response: The suggested changes have been made (page 10, line 158).

The definition of mediator factor is a variable that is in the middle of the causal path of exposure and outcome. SGA birth is not a cause of sex, for example. So, please check the classification of preterm birth and sex as mediators in the models.

Response: We rephrased the section to clarify the exact analyses performed, indicating the exposure and outcomes clearly, with the mediators and confounders in each analysis (page 10; lines 161,162, page 11, lines 166, 167, 171). Also, we fully reviewed the results section and made the necessary changes regarding the use of the word “mediator” or “confounder”.

Please check table 3 adjusted value of SGA for In-hospital mortality, the RR change from 7.95 (4.7– 13.46) to 226.56 (3.47– 318.22). Is it right? Which variables are causing this change?

Response: The indicated adjusted RR (226.56 (3.47– 318.22), is after adjusting for the risk variables associated with SGA that were significant in the univariate analysis, with p-values <0.05, presented in Table 2, which are as follows: gestational age, maternal age, parity, nationality, education, diabetes status, chronic hypertension, early- or pre-pregnancy BMI, baby gender, and any chromosomal or congenital abnormalities (Table 3–A). The number of SGA who died in the hospital is very small (n: 22/882 (2.5%), and thus after adjusting for other risk variables, the RR became overestimated. We have indicated this as one of the study limitations in the discussion (Page 36, Line 465-467)

Reviewers' comments:

Reviewer #1:

- It is not clear which standardized birth weight chart was used to classify neonates into SGA, AGA and LGA. Why using six methods (charts) based on fetal growth/weight standards for classifying appropriateness of birth weight?

Response: We have used a tool encompassing 6 fetal growth standards, among which we chose the INTERGROWTH-21st birth weight standard to calculate the percentiles. We have added a section in the introduction to explain in detail why we based our calculations on this particular reference (Pages 6,7; Lines 66-77). Also, we clarified it in the methods (Page 8, Lines 108-113).

- Surprisingly, the method used for classifying adequacy of birth weight resulted in 6% of SGA and 16% of LGA neonates. Considering that the expected rate would be 10% for SGA and LGA, why did you find such discrepancy? What are the implications for 1) the interpretation of your findings and 2) the generalizability of your study?

Response: We have added a section to discuss the comments indicated regarding the observed disparities, implications, interpretation, and generalizability (Pages 31,32, Lines 363-395).

- What contribution to clinical practice does your study give?

Response: We have added a section to further highlight the contribution of the study findings to clinical practice (page 36, lines 473-485).

Reviewer #2: Thanks for the opportunity to re-review this manuscript where the aim was to assess the incidence, risk factors, and feto-maternal outcomes associated with small-forgestational age (SGA) and large-for-gestational age (LGA) infants.

The comments have been adressed, however it is still confusing and un-clear with the exposure and outcome in the different analyses. You write:“Firstly, logistic regression analysis was performed for risk factors/confounders (demographic and medical factors) and mediators (prematurity and gender) of appropriateness of fetal growth for the GA groups (SGA/LGA vs. AGA).”

As I understand it you have used SGA/LGA as outcome in these analysis and different risk factors as exposure. Which variables are confounders/mediators will depend on which variable is exposure and which one is the outcome. If SGA/LGA is the outcome preterm birth will be a mediator. However, in the analyses of outcome: “Secondly, logistic regression was performed to investigate the outcomes of SGA and LGA including Apgar score, NICU/death in LR/OT, and in-hospital mortality. Multiple logistic regression was performed (including all significant confounders and mediators from the univariate analysis) to investigate the association of SGA/LGA with Apgar score, NICU/death in LR/OT, and in-hospital mortality as outcomes.” I guess that in these analyses SGA/LGA were the exposure and then preterm birth has another role, in this setting preterm birth will be a confounder (or in some cases a mediator if we believe that the SGA/LGA caused the premature birth) Gender will be a mediator in the first analyses and a confounder in the second analyses. Please, make it clear what the exposure and the outcome is in the different analyses.

Response: We agree and thus we rephrased the section to clarify the exact analyses performed, indicating the exposure and outcomes clearly, with the mediators and confounders in each analysis. We agree that both gender and preterm birth are confounders in the second analysis and not mediators, thus we deleted the word “mediators” from the second analysis (pages 10, 11; lines 161, 173), also, we fully reviewed the results section and made the necessary changes regarding the use of the word “mediator” or “confounder”. 

I note that attributable fractions are added. For example you write “not delivered preterm 11.6%, indicating that almost 12% of SGA cases could have been prevented if mothers had not delivered preterm”, it would be good to add, in the discussion part, something about possible unmeasured confounding.

Response: We have added possible unmeasured confounding in the discussion (page 35, lines 459-462). 

- The numbers in table 1, “pregnancy mode” look strange. The percentages do not add up to 100%.

Response: We agree with the reviewer, there was an error in the table, the table has been fully reviewed, and the error has been fixed (Table 1).

- In the conculsion you write: "SGA and LGA births are related multi-factor interactions of demographic and medical confounders that can be mediated by prematurity and gender of the baby". This sentence is un-clear. What do you mean? What do you mean with interactions?

Response: We do agree with the reviewer that the sentence is unclear. The indicated sentence has been rephrased (pages 36; lines 475-485).

---

## [Editor Report · Decision Letter 2]

11 Oct 2021

Incidence, Risk Factors, and Feto-Maternal Outcomes of Inappropriate Birth Weight for Gestational Age Among Singleton Live Births in Qatar: A Population-Based Study

PONE-D-21-07926R2

Dear Dr. Al-Dewik,

We’re pleased to inform you that your manuscript has been judged scientifically suitable for publication and will be formally accepted for publication once it meets all outstanding technical requirements.

Kind regards,

Dayana Farias, Ph.D

Academic Editor

PLOS ONE

---

## [Editor Report · Acceptance letter]

15 Oct 2021

PONE-D-21-07926R2 

Incidence, Risk Factors, and Feto-Maternal Outcomes of Inappropriate Birth Weight for Gestational Age Among Singleton Live Births in Qatar: A Population-Based Study 

Dear Dr. Al-Dewik:

I'm pleased to inform you that your manuscript has been deemed suitable for publication in PLOS ONE. Congratulations! Your manuscript is now with our production department. 

Kind regards, 

on behalf of

Dr. Dayana Farias 

Academic Editor

PLOS ONE